# BRCA1 preserves genome integrity during the formation of undifferentiated spermatogonia

Peng Li [ID][1,2,7], Licun Song [ID][1,3,7], Longfei Ma[4], Chunsheng Han [ID][4], Lejun Li [ID][1,2✉], Lin-Yu Lu [ID][1,3,5✉] & Yidan Liu [ID][1,6✉]

## Abstract

**Undifferentiated spermatogonia, which form shortly after birth, consist of spermatogonial stem cells and progenitor spermatogonia that maintain homeostasis. As the origin of spermatogenesis, undifferentiated spermatogonia must preserve genome integrity. Paradoxically, we demonstrate that massive spontaneous DNA damage, potentially generated by formaldehyde, arises during the formation of undifferentiated spermatogonia, posing a significant threat to genome integrity. We further reveal that BRCA1 is essential for the timely repair of this spontaneous DNA damage. BRCA1 loss leads to a dramatic reduction in progenitor spermatogonia and disrupts the formation of undifferentiated spermatogonia. Although spermatogonial stem cells initially undergo hyperproliferation, they are eventually depleted, resulting in the premature exhaustion of undifferentiated spermatogonia. Our study highlights a striking difference in DNA damage sensitivity between the two populations of undifferentiated spermatogonia and underscores the critical role of BRCA1-dependent DNA damage repair in preserving genome integrity during the formation of undifferentiated spermatogonia.**

**Keywords** Undifferentiated Spermatogonia; BRCA1; DNA Damage Repair; Genome Integrity
**Subject Categories** Development; DNA Replication, Recombination & Repair; Stem Cells & Regenerative Medicine

## Introduction

In mammals, spermatogenesis proceeds continuously to sustain long-term male fertility. Undifferentiated spermatogonia, which form shortly after birth, serve as the origin of all germ cells in spermatogenesis. These cells, also known as type A spermatogonia in mice, are a rare population located at the basal lamina of the seminiferous tubules in the testes (Kanatsu-Shinohara and Shinohara, 2013; Oatley and Brinster, 2012). Like other tissue-specific stem cells, undifferentiated spermatogonia can either self-renew to maintain their population or transition into differentiating spermatogonia, which are committed to meiosis (Griswold, 2016). Exhaustion of undifferentiated spermatogonia results in the complete loss of germ cells in the testes and leads to male infertility.

Studies have revealed that undifferentiated spermatogonia in mice consist of a mixture of self-renewing spermatogonial stem cells and differentiation-primed progenitor spermatogonia, each exhibiting distinct gene expression profiles (Makela and Hobbs, 2019; Yoshida, 2019). Spermatogonial stem cells, which are positive for GFRα1, maintain the essential stem cell properties required for long-term spermatogenesis (He et al, 2007). In contrast, progenitor spermatogonia, which express SOX3, NGN3, and RARγ, possess the potential to initiate differentiation upon retinoic acid stimulation (Ikami et al, 2015; Nakagawa et al, 2010; Suzuki et al, 2012). Although spermatogonial stem cells and progenitor spermatogonia exhibit distinct characteristics, these two populations can interconvert to maintain the homeostasis of undifferentiated spermatogonia in adult mice (Hara et al, 2014; Nakagawa et al, 2010).

Since the integrity of the genome, which will be passed on to offspring, must be carefully preserved, undifferentiated spermatogonia are extremely sensitive to DNA damage, as it poses an enormous threat to genome integrity. These cells are rapidly eliminated in mice upon exposure to exogenous DNA damage, such as that caused by alkylating agents (La and Hobbs, 2019; Zohni et al, 2012). On the other hand, spontaneous DNA damage also arises during normal cellular processes, making robust DNA damage repair mechanisms essential for undifferentiated spermatogonia to efficiently manage endogenous DNA damage. Consistent with this idea, the removal of critical DNA damage repair proteins in mice disrupts the maintenance of undifferentiated spermatogonia and leads to male infertility (Takubo et al, 2008; Tang et al, 2024).

As a key player in DNA damage repair in somatic cells, BRCA1 is indispensable for spermatogenesis and male fertility in mice (Broering et al, 2014; Shakya et al, 2011; Turner et al, 2004; Xu et al, 2003). In addition to its well-characterized role in promoting

[1]Key Laboratory of Reproductive Genetics (Ministry of Education), Women's Hospital, Zhejiang University School of Medicine, Hangzhou, China. [2]Zhejiang Key Laboratory of Maternal and Infant Health, Women's Hospital, Zhejiang University School of Medicine, Hangzhou, China. [3]Zhejiang Key Laboratory of Frontier Medical Research on Cancer Metabolism, Institute of Translational Medicine, Zhejiang University School of Medicine, Hangzhou, China. [4]State Key Laboratory of Stem Cell and Reproductive Biology, Institute of Zoology, Chinese Academy of Sciences, Beijing, China. [5]Cancer Center, Zhejiang University, Hangzhou, China. [6]Zhejiang Key Laboratory of Precision Diagnosis and Therapy for Major Gynecological Diseases, Women's Hospital, Zhejiang University School of Medicine, Hangzhou, China. [7]These authors contributed equally: Peng Li, Licun Song.
✉E-mail: lilejun@zju.edu.cn; lulinyu@zju.edu.cn; yidanliu@zju.edu.cn

meiotic sex chromosome inactivation (MSCI) (Broering et al, 2014; Turner et al, 2004), we have recently demonstrated that BRCA1 is also essential for meiotic recombination (Bai et al, 2024). Besides its critical function in meiotic prophase, it remains unclear whether BRCA1 is required to maintain undifferentiated spermatogonia. No defects in undifferentiated spermatogonia have been reported in mice carrying *Brca1* hypomorphic allele (Broering et al, 2014; Cressman et al, 1999; Xu et al, 2003). However, since BRCA1's functions are partially retained in these mice, germ cell-specific *Brca1* null mice are needed to clarify the role of BRCA1 in undifferentiated spermatogonia.

Although DNA damage repair deficiency disrupts the long-term maintenance of undifferentiated spermatogonia in adult mice, little is known about its impact on the formation of undifferentiated spermatogonia, during which spermatogonial stem cells are converted to progenitor spermatogonia until the homeostasis between these two populations is established. Given that undifferentiated spermatogonia are formed shortly after birth, endogenous DNA damage accumulated during this stage might be expected to be too low to threaten genomic stability. Unexpectedly, however, we demonstrate in this study that massive spontaneous DNA damage arises during the formation of undifferentiated spermatogonia. Using germ cell-specific *Brca1* null mice, we reveal an indispensable role of BRCA1-dependent DNA damage repair in preserving genome integrity during this process. BRCA1 loss differentially impacts spermatogonial stem cells and progenitor spermatogonia, ultimately leading to the premature exhaustion of undifferentiated spermatogonia.

## Results

### Spontaneous DNA damage arises during the formation of undifferentiated spermatogonia

Undifferentiated spermatogonia are formed after birth (Fig. EV1A). Gonocytes are precursors of undifferentiated spermatogonia (Culty, 2009). At embryonic day (E) 18.5, the transition from gonocytes to undifferentiated spermatogonia has not started in wild-type (WT) mice yet. All cells were positive for germ cell marker GCNA but were negative for PLZF, a marker for undifferentiated spermatogonia (Fig. EV1B). At postnatal day (PD) 1, PLZF-positive cells could be observed in most GCNA-positive germ cells in WT mice (Fig. EV1B), suggesting that undifferentiated spermatogonia start to form. At this stage, all undifferentiated spermatogonia were positive for GFRα1, a marker for the spermatogonial stem cells, but were negative for SOX3, a marker for the progenitor spermatogonia (Fig. EV1C,D). At PD4, SOX3-positive progenitor spermatogonia started to be observed. The percentage of SOX3-positive progenitor spermatogonia gradually increased, and the percentage of GFRα1-positive spermatogonial stem cells gradually decreased at PD4, PD7, and PD14, but the percentages of these two populations were similar between PD14 and PD21 (Fig. EV1C–E). Therefore, the undifferentiated spermatogonia are fully formed in WT mice at PD14, when the homeostasis between spermatogonial stem cells and progenitor spermatogonia has been established.

To examine if DNA damage repair is required during the formation of undifferentiated spermatogonia, we first examined whether DNA damage arises during this process in WT mice. At

E18.5, no γH2AX signals could be observed in the gonocytes (Fig. 1A), indicating no DNA damage. Interestingly, strong γH2AX signals were observed in PLZF-positive germ cells during the formation of undifferentiated spermatogonia at PD1, PD4, and PD7 (Fig. 1A,C). Consistent with the presence of DNA damage, both pan-nuclear γH2AX signals and γH2AX foci were observed in these cells (Fig. EV2A,B). When undifferentiated spermatogonia were fully formed at PD14, γH2AX signals could no longer be observed in PLZF-positive germ cells (Fig. 1A,C) but were found exclusively in spermatocytes at meiotic prophase. On the contrary, no γH2AX signals could be observed in SOX9-positive Sertoli cells from PD1 to PD14 (Fig. 1B,C). Importantly, the γH2AX signals were specific since their signals could not be observed during the formation of undifferentiated spermatogonia in *H2ax* KO mice (Fig. EV2C,D). Although the spermatogonial differentiation begins in WT mice at PD4, only around 20% of γH2AX-positive cells were positive for KIT, a marker for differentiating spermatogonia (Busada et al, 2015), suggesting that spermatogonial differentiation is unlikely the reason for the γH2AX signals in PLZF-positive germ cells (Fig. EV2E,F). These observations collectively demonstrate that strong DNA damage arises specifically during the formation of undifferentiated spermatogonia. Interestingly, despite the presence of strong DNA damage in undifferentiated spermatogonia, no cells positive for cleaved PARP1 were observed in testes of WT mice at PD4 or PD7 (Fig. 1D), suggesting that the DNA damage was repaired efficiently so that no apoptotic cell death occurred.

### BRCA1 is required for early postnatal germ cell development

The observations above suggest that robust DNA damage repair exists during the formation of undifferentiated spermatogonia, but the underlying mechanism is unclear. A previous study of gene expression profiles of undifferentiated spermatogonia in infant and adult mice has revealed that the expressions of genes in BRCA1-related DNA damage response pathways are higher in PD6 than in adults (Hermann et al, 2018), suggesting that BRCA1 may be a key DNA damage repair protein during the formation of undifferentiated spermatogonia. To examine if BRCA1 is required for DNA damage repair during the formation of undifferentiated spermatogonia, we generated *Brca1* germ cell-specific knockout (KO) mice (*Brca1^f/-^ Vasa-Cre* mice, from now on referred to as *Brca1* vKO mice) using *Vasa-Cre* transgenic mice (Fig. EV3A–C) (Gallardo et al, 2007). The *Vasa-Cre* transgene starts to express in gonocytes at E15.5 (Fig. EV3A), and Cre-mediated deletion of exon 5–13 of *Brca1* completely disrupts the expression of BRCA1 when undifferentiated spermatogonia started to form at PD1 (Fig. 2A), making *Brca1* vKO a suitable mouse model to study the impact of BRCA1 loss on the formation of undifferentiated spermatogonia.

Although the testes of *Brca1* vKO male mice were indistinguishable from those of control mice at PD1, they were significantly smaller than those of control mice starting from PD7 (Figs. 2B,C and EV3D,E), suggesting that BRCA1 loss likely affects early postnatal germ cell development before PD7. Given that undifferentiated spermatogonia formation is not yet complete at PD4 but nearly finished by PD7 in WT mice, PD7 is the optimal timepoint to analyze the impact of BRCA1 loss on undifferentiated spermatogonia formation. To analyze the defects in *Brca1* vKO testes comprehensively, we isolated testes from one pair of control

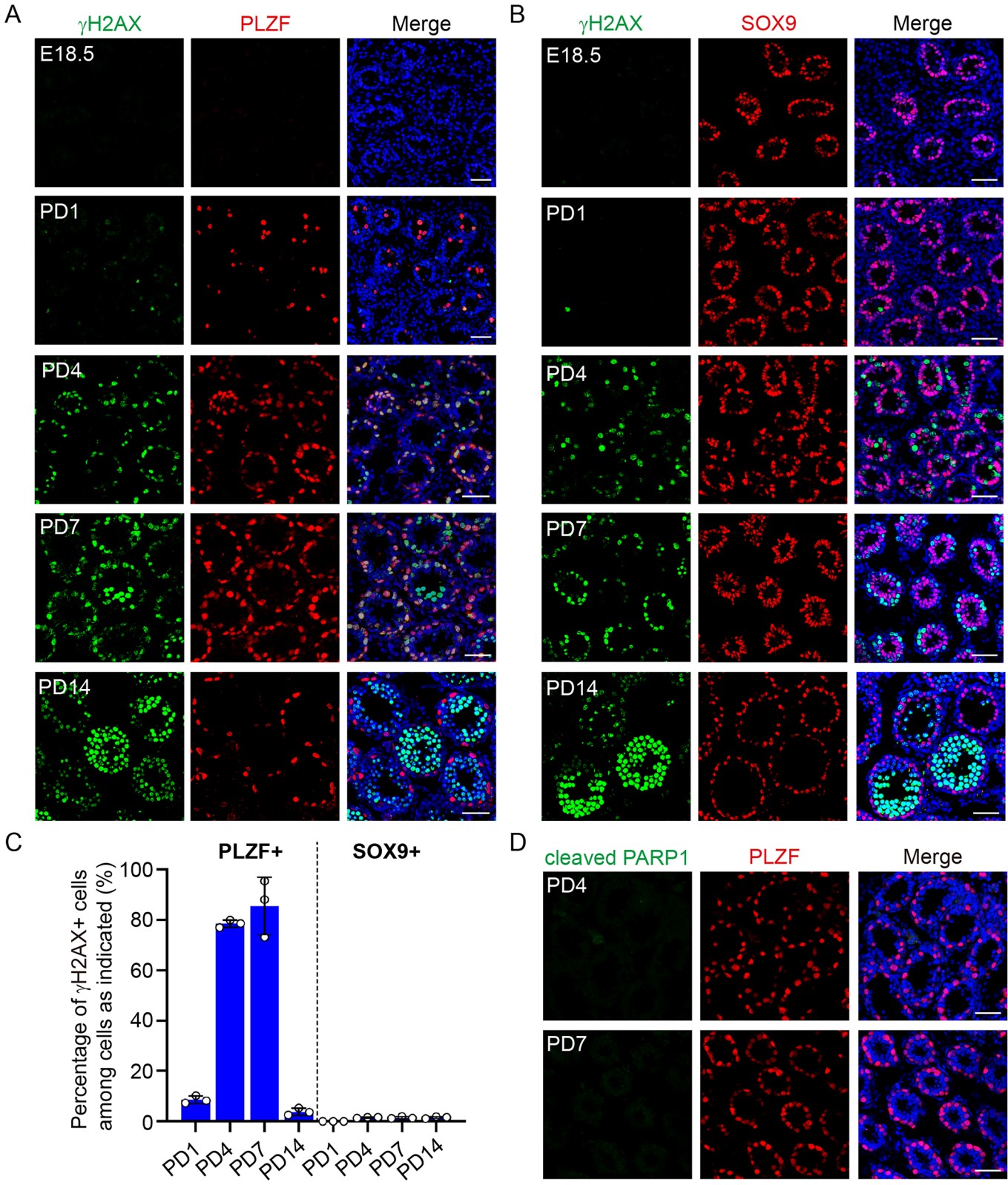

and *Brca1* vKO male mice at PD7 and performed single-cell RNA sequencing (scRNA-seq) analyses using the 10× Genomics platform. Since no biological repeats were included in this pilot experiment, potentially causing problems in data interpretation, the

scRNA-Seq analyses were subsequently validated using immunofluorescence staining at different developmental stages.

After filtering low-quality cells, 5077 cells from control testes and 8711 cells from *Brca1* vKO testes were retained for downstream

**Figure 1. Spontaneous DNA damage arises during the formation of undifferentiated spermatogonia.**

(A) Immunofluorescence (IF) staining of γH2AX and PLZF in frozen sections of testes from WT mice at E18.5, PD1, PD4, PD7, and PD14. (B) IF staining of γH2AX and SOX9 in frozen sections of testes from WT mice at E18.5, PD1, PD4, PD7, and PD14. (C) Statistical analysis of the percentage of γH2AX+ cells among PLZF+ cells and SOX9+ cells in WT testes at PD1, PD4, PD7, and PD14. At least 100 PLZF+ cells or SOX9+ cells from each mouse were analyzed. (D) IF staining of cleaved PARP1 and PLZF in frozen sections of testes from WT mice at PD4 and PD7. Data information: Data are presented as mean ± SD. Three mice at each timepoint were analyzed. Scale bar, 50 μm. Source data are available online for this figure.

analysis. Unsupervised clustering of 13,788 cells was conducted using t-distributed stochastic neighbor embedding (tSNE) analysis. Eight major clusters were identified, including one germ cell and seven somatic populations (Fig. 2D–F). The identity of each cluster was confirmed by known cell type markers reported from previous studies (Green et al, 2018; Wang et al, 2019). All clusters were present in both control and *Brca1* vKO testes, but the percentage of germ cell population significantly decreased in *Brca1* vKO testes (Fig. 2G). To verify the result of scRNA-seq, we performed immunofluorescent staining of germ cell marker MVH in control and *Brca1* vKO testes. The numbers of germ cells were similar between control and *Brca1* vKO testes at PD1 but were dramatically reduced in *Brca1* vKO testes at PD7 (Fig. 2H,I). Therefore, BRCA1 loss indeed disrupts early postnatal germ cell development.

## Progenitor spermatogonia are more susceptible to BRCA1 loss

To examine the cause of germ cell depletion in *Brca1* vKO testes, we further analyzed the germ cell populations in scRNA-seq data. We re-clustered the 1257 cells from the germ cell population using tSNE, and marker gene expression analysis was conducted to confirm cell types. Three distinct clusters in a possible developmental order were identified (Figs. 3A and EV4A), which were spermatogonial stem cells (SPG1), progenitor spermatogonia (SPG2), and differentiating spermatogonia and early meiotic cells (SPG3), respectively. We further compared the percentages of each cluster in all testicular cells between control and *Brca1* vKO testes. Consistent with the overall reduction in the germ cell population, the percentages of all three clusters were reduced in *Brca1* vKO testes, but the reduction was more severe in SPG2 and SPG3 than in SPG1 (Fig. 3B). To examine which step of germ cell development was most affected in *Brca1* vKO testes, we calculated the percentage ratio between adjacent clusters. The percentage ratio between SPG2 and SPG1 in *Brca1* vKO testes was less than 1/3 of that in control testes, and the percentage ratio between SPG3 and SPG2 in *Brca1* vKO testes was 1.6-fold of that in control testes (Fig. 3C). Therefore, the germ cell development from SPG1 to SPG2 was most affected in *Brca1* vKO testes. These observations also revealed that SPG2 was much more susceptible to BRCA1 loss than SPG1, and the susceptibility was not further increased in SPG3.

To verify if progenitor spermatogonia (SPG2) are more susceptible to BRCA1 loss than spermatogonial stem cells (SPG1), we utilized immunofluorescent staining to confirm the identity of the remaining germ cells in *Brca1* vKO testes at PD7. Most remaining germ cells in *Brca1* vKO testes were PLZF-positive undifferentiated spermatogonia at PD7, but they were significantly decreased compared with those in control testes (Figs. 3D and EV4B–D). Consistent with the scRNA-seq data, the numbers of GFRα1-positive spermatogonial stem cells and SOX3-positive

progenitor spermatogonia were both reduced in *Brca1* vKO testes, but the reduction was much more dramatic in SOX3-positive progenitor spermatogonia (Figs. 3E and EV4B–D). Among all PLZF-positive undifferentiated spermatogonia, the percentage of GFRα1-positive spermatogonial stem cells significantly increased, and that of SOX3-positive progenitor spermatogonia dramatically decreased in *Brca1* vKO testes (Fig. 3F). As a result, there was a dramatic change in the composition of PLZF-positive undifferentiated spermatogonia in *Brca1* vKO testes, with GFRα1-positive spermatogonial stem cells being the predominant population (Fig. 3F). These observations suggest that progenitor spermatogonia are more susceptible to BRCA1 loss than spermatogonial stem cells during the formation of undifferentiated spermatogonia.

To analyze if gene expression changes caused by BRCA1 loss contribute to the phenotypes observed in *Brca1* vKO testes, we conducted differential gene expression analysis for the three cell clusters and performed pathway enrichment analysis on the up- and down-regulated genes (Appendix Fig. S1). Since the germ cell development from SPG1 to SPG2 was most affected in *Brca1* vKO testes, we focused on pathways enriched in these two clusters. However, we did not identify the enrichment of DNA damage checkpoint or relevant pathways in SPG1 or SPG2, suggesting that gene expression changes caused by BRCA1 loss are unlikely to contribute directly to the phenotypes in *Brca1* vKO testes.

## Spermatogonial stem cells hyperproliferate after BRCA1 loss

In control testes, the homeostasis of undifferentiated spermatogonia was almost established at PD7. Due to the dramatic reduction of SOX3-positive progenitor spermatogonia, the homeostasis of undifferentiated spermatogonia failed to be established in *Brca1* vKO testes at PD7. Interestingly, although the number of GFRα1-positive spermatogonial stem cells was also decreased in *Brca1* vKO testes at PD7, the percentage of p-H3(S10)-positive mitotic cells in this population was much higher than that in control testes (Fig. 4A,B). This observation reveals an unexpected hyperproliferation of GFRα1-positive spermatogonial stem cells, which could be interpreted as an attempt to restore the homeostasis of undifferentiated spermatogonia. Therefore, we examined if the homeostasis of undifferentiated spermatogonia could be gradually established in older *Brca1* vKO mice.

In control testes, the number of GFRα1-positive spermatogonial stem cells decreased to a level at PD14 that remained unchanged at PD21. In *Brca1* vKO testes, although the number of GFRα1-positive spermatogonial stem cells was reduced at PD7, it did not further decrease but instead increased at PD14 and continued to increase to a level 2–3 fold of that in control testes at PD21 (Figs. 4C,D and EV5A,B). Consistently, the percentage of p-H3(S10)-positive mitotic cells in GFRα1-positive spermatogonial stem cells remained

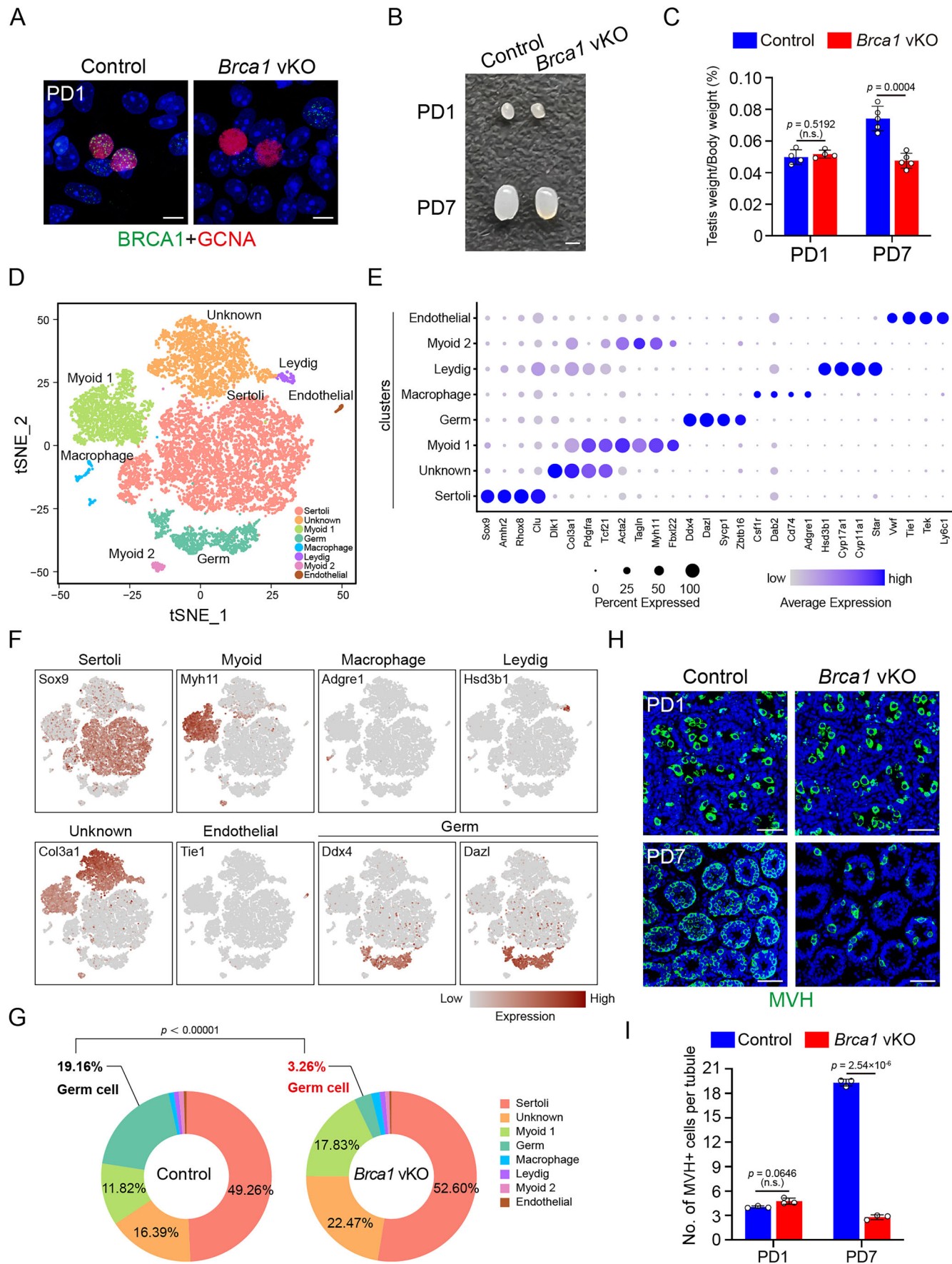

◄ **Figure 2. BRCA1 is required for early postnatal germ cell development.**

(A) IF staining of BRCA1 and GCNA in testicular cells of Control and *Brca1* vKO male mice at PD1 after 10 Gy IR treatment. Scale bar, 10 μm. (B) Representative images of testes from Control and *Brca1* vKO male mice at PD1 and PD7. Scale bar, 1 mm. (C) Ratios of testis weight to body weight of Control and *Brca1* vKO male mice at PD1 and PD7. At least four mice of each genotype were analyzed. (D) tSNE and clustering analysis of combined single-cell transcriptome data from testicular cells of Control and *Brca1* vKO male mice at PD7. Each dot represents a single cell and cell clusters are distinguished by different colors. (E) Dot plot for expression of selected marker genes in each cell type. (F) Gene expression patterns of selected marker genes projected on the tSNE plots. (G) Pie diagram to show the distribution of each cell type among all testicular cells of Control and *Brca1* vKO male mice at PD7. The percentage of germ cells in two groups was highlighted. The *P* value for the percentage of germ cells in control and *Brca1* vKO groups was calculated using the chi-square test, yielding a *P* value of <0.00001. (H) IF staining of MVH in frozen sections of testes from Control and *Brca1* vKO male mice at PD1 and PD7. Scale bar, 50 μm. (I) Statistical analysis of the number of MVH+ cells per seminiferous tubule in testes of Control and *Brca1* vKO male mice at PD1 and PD7. Three mice of each genotype were analyzed. Data information: Data are presented as mean ± SD. *P* value, two-tailed unpaired Student's *t* test. n.s. not significant. Source data are available online for this figure.

higher than that in control testes at PD14 and PD21, indicative of continued hyperproliferation in these cells (Appendix Fig. S2). On the contrary, the number of SOX3-positive progenitor spermatogonia only slightly increased at PD14 and PD21 (Figs. 4E,F and EV5C,D). Among all PLZF-positive undifferentiated spermatogonia in *Brca1* vKO testes at PD14 and PD21, the percentage of GFRα1-positive spermatogonial stem cells remained much higher than that in control testes, and the percentage of SOX3-positive progenitor spermatogonia remained much lower than that in control testes (Fig. 4G), suggesting that the composition of PLZF-positive undifferentiated spermatogonia was not restored in *Brca1* vKO testes. Therefore, despite the hyperproliferation of spermatogonial stem cells, the homeostasis of undifferentiated spermatogonia was never established in *Brca1* vKO testes.

## Gradual loss of spermatogonial stem cells leads to premature exhaustion of undifferentiated spermatogonia in *Brca1* vKO testes

Most spermatogonial stem cells are usually in a quiescent state, which is essential for maintaining spermatogenesis for many years. To explore the consequences of hyperproliferation of spermatogonial stem cells, we examined the fate of these cells in older *Brca1* vKO mice. Although the number of GFRα1-positive spermatogonial stem cells in *Brca1* vKO testes was 2–3 times that in control testes at PD21, it rapidly decreased to a level slightly more than that in control testes at PD42 and to a level similar to that in control testes at PD90 (Fig. 5A,B). At PD180, GFRα1-positive spermatogonial stem cells were hardly detectable in *Brca1* vKO testes (Fig. 5A,B). Consistent with GFRα1-positive spermatogonial stem cells being the predominant population of undifferentiated spermatogonia due to the dramatic reduction of SOX3-positive progenitor spermatogonia, the number of PLZF-positive undifferentiated spermatogonia also rapidly decreased, and they were hardly detectable at PD180 in *Brca1* vKO testes (Fig. 5A,C). In line with these observations, the number of germ cells quickly reduced, and the seminiferous tubules were almost completely devoid of germ cells and contained many vacuoles at PD180 in *Brca1* vKO testes (Fig. 5D). Therefore, accompanied by the gradual loss of spermatogonial stem cells, the undifferentiated spermatogonia were exhausted prematurely in *Brca1* vKO testes.

## Blocking apoptotic cell death of progenitor spermatogonia restores the homeostasis of undifferentiated spermatogonia in *Brca1* vKO testes

In *Brca1* vKO testes, the dramatic reduction of progenitor spermatogonia seems to be the most upstream event that disrupts

the establishment of homeostasis of undifferentiated spermatogonia, triggers hyperproliferation of spermatogonial stem cells, and leads to premature exhaustion of undifferentiated spermatogonia. To further interrogate this idea, we examined apoptosis in these cells. Little apoptosis was observed in control testes at PD7, PD14, and PD21 (Appendix Fig. S3A–C). In *Brca1* vKO testes, the number of SOX3-positive progenitor spermatogonia was dramatically reduced at PD7, and little apoptosis was observed in the remaining cells (Appendix Fig. S3A–C). This suggests that the rapid elimination of SOX3-positive progenitor spermatogonia likely prevents us from capturing cells undergoing apoptosis at PD7. Similarly, little apoptosis was observed in GFRα1-positive spermatogonial stem cells in *Brca1* vKO testes at PD7 (Appendix Fig. S3A–C), likely because the detrimental effects of hyperproliferation had not yet manifested at this early stage.

By PD14 and PD21, as the number of remaining SOX3-positive progenitor spermatogonia increased in *Brca1* vKO testes, apoptosis became detectable (Appendix Fig. S3A–C). Concurrently, as the hyperproliferation of GFRα1-positive spermatogonial stem cells continued, apoptosis was also observed in this population of cells (Appendix Fig. S3A–C). Importantly, apoptosis was much more pronounced in SOX3-positive progenitor spermatogonia than in GFRα1-positive spermatogonial stem cells (Appendix Fig. S3C). These observations are consistent with our conclusions that progenitor spermatogonia are more susceptible to BRCA1 loss than spermatogonial stem cells, and the dramatic reduction of progenitor spermatogonia seems to be the most upstream event that disrupts the establishment of homeostasis of undifferentiated spermatogonia.

To test if blocking apoptotic cell death of progenitor spermatogonia could restore the homeostasis of undifferentiated spermatogonia and prevent the hyperproliferation of spermatogonial stem cells in *Brca1* vKO testes, we generated *Brca1* and *p53* germ cell-specific double knockout mice (*Brca1^f/-^ Trp53^f/-^ Vasa-Cre* mice, from now on referred to as *Brca1-p53* vDKO mice). Unlike in *Brca1* vKO testes, SOX3-positive progenitor spermatogonia were readily observed in *Brca1-p53* vDKO testes at PD7 (Fig. 6B). Although still lower than in control testes, the number of SOX3-positive progenitor spermatogonia in *Brca1-p53* vDKO testes was significantly higher than that in *Brca1* vKO testes at PD7 (Fig. 6C). Consistent with our hypothesis, the reduction of GFRα1-positive spermatogonial stem cells was not observed in *Brca1-p53* vDKO testes at PD7 either (Fig. 6A,C). In addition, although the number of PLZF-positive undifferentiated spermatogonia remained lower (Fig. 6C), the composition of PLZF-positive undifferentiated spermatogonia in *Brca1-p53* vDKO testes was similar to that in control testes (Fig. 6D). Therefore, the homeostasis of undifferentiated spermatogonia was largely restored in *Brca1-p53* vDKO testes at PD7.

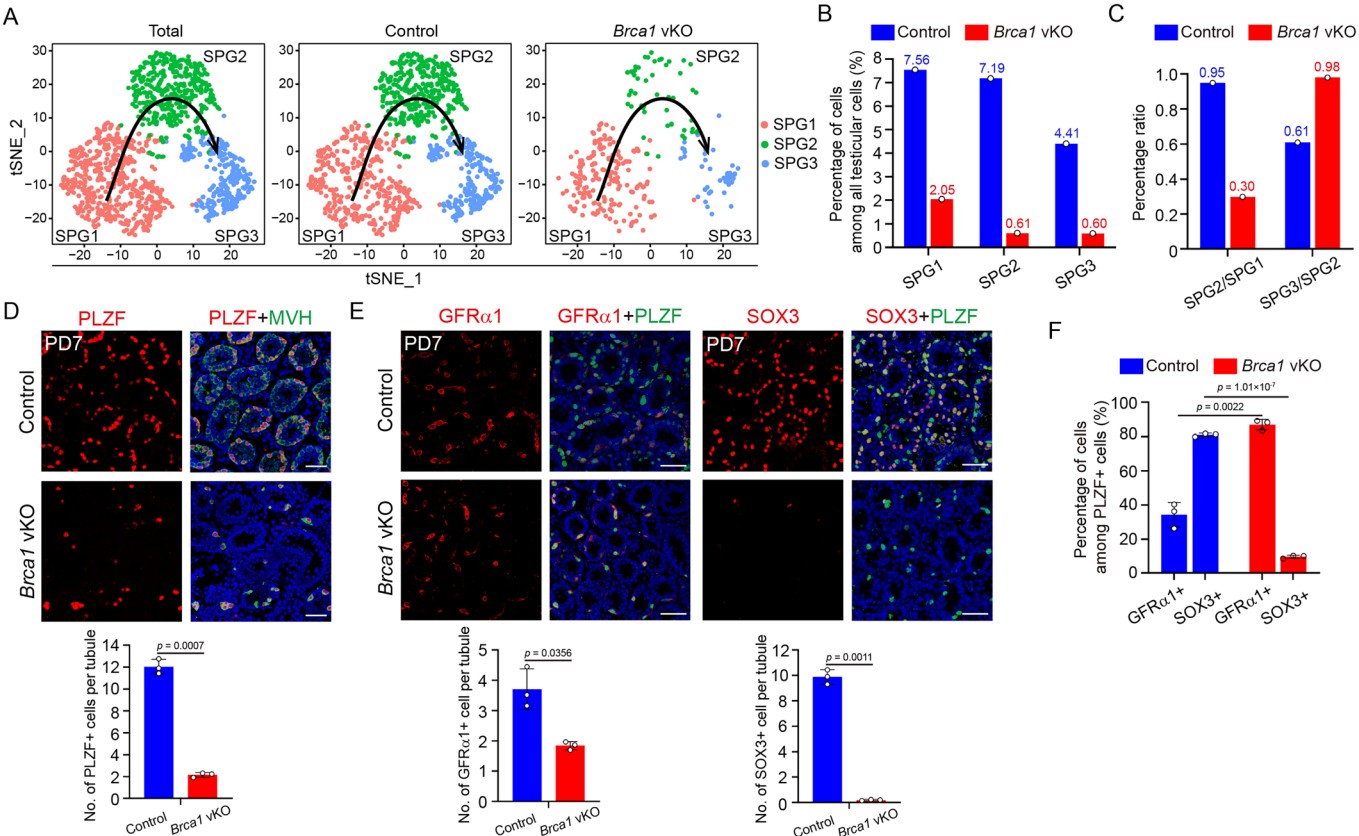

**Figure 3. Progenitor spermatogonia are more susceptible to BRCA1 loss.**

(A) Clustering of germ cell population based on tSNE analysis identifies three major subtypes: SPG1 to SPG3. The arrow indicates the stages of development from SPG1 to SPG3. (B) Comparative analysis of the percentage of SPG1, SPG2, and SPG3 among all testicular cells between Control and *Brca1* vKO male mice at PD7. (C) Analysis of the percentage ratio of SPG2/SPG1 and SPG3/SPG2 in Control and *Brca1* vKO male mice at PD7. (D) IF staining of PLZF and MVH in frozen sections of testes from Control and *Brca1* vKO male mice at PD7 and statistical analysis of the number of PLZF+ cells per seminiferous tubule. (E) IF staining of GFRα1&PLZF and SOX3&PLZF in frozen sections of testes from Control and *Brca1* vKO male mice at PD7 and statistical analysis of the number of GFRα1+ and SOX3+ cells per seminiferous tubule. (F) Statistical analysis of the percentage of GFRα1+ and SOX3+ cells among PLZF+ cells in testes of Control and *Brca1* vKO male mice. Data information: Data are presented as mean ± SD. Three mice of each genotype were analyzed. *P* value, two-tailed unpaired Student's *t* test. Scale bar, 50 μm. Source data are available online for this figure.

At PD21, the numbers of SOX3-positive progenitor spermatogonia, GFRα1-positive spermatogonial stem cells, and PLZF-positive undifferentiated spermatogonia in *Brca1-p53* vDKO testes were similar to those in control testes, and the composition of PLZF-positive undifferentiated spermatogonia in *Brca1-p53* vDKO testes was similar to that in control testes (Fig. 6E–H). Therefore, accompanied by the restoration of the homeostasis of undifferentiated spermatogonia, the hyperproliferation of spermatogonial stem cells was also reversed in *Brca1-p53* vDKO testes at PD21 (Fig. 6E,G). Taken together, these observations support the idea that a dramatic reduction of progenitor spermatogonia disrupts the establishment of homeostasis of undifferentiated spermatogonia, triggers hyperproliferation of spermatogonial stem cells, and leads to premature exhaustion of undifferentiated spermatogonia in *Brca1* vKO testes.

## Rapid proliferation of progenitor spermatogonia potentially contributes to their susceptibility to BRCA1 loss

Since the dramatic reduction of progenitor spermatogonia triggers the defects in *Brca1* vKO testes, we examined why progenitor

spermatogonia are more susceptible to BRCA1 loss than spermatogonial stem cells. At PD7, strong γH2AX signals were observed in GFRα1-positive spermatogonial stem cells and SOX3-positive progenitor spermatogonia in both control and *Brca1* vKO testes (Fig. 7A). The intensities of γH2AX signals and the percentages of γH2AX-positive undifferentiated spermatogonia were similar between control and *Brca1* vKO testes (Fig. 7B,C), regardless of GFRα1-positive spermatogonial stem cells and SOX3-positive progenitor spermatogonia analyzed, suggesting BRCA1 loss has no impact on the generation of spontaneous DNA damage. Importantly, the intensities of γH2AX signals and the percentages of γH2AX-positive cells were similar between both populations in control and *Brca1* vKO testes (Fig. 7B,C), suggesting that spontaneous DNA damage arises similarly in these two populations.

At PD14 and PD21, the γH2AX signals diminished in both GFRα1-positive spermatogonial stem cells and SOX3-positive progenitor spermatogonia in control testes (Fig. 7D–I), suggesting that the DNA damage repair dynamics are similar between these two populations in control testes. In contrast, the γH2AX signals in GFRα1-positive spermatogonial stem cells and SOX3-positive progenitor

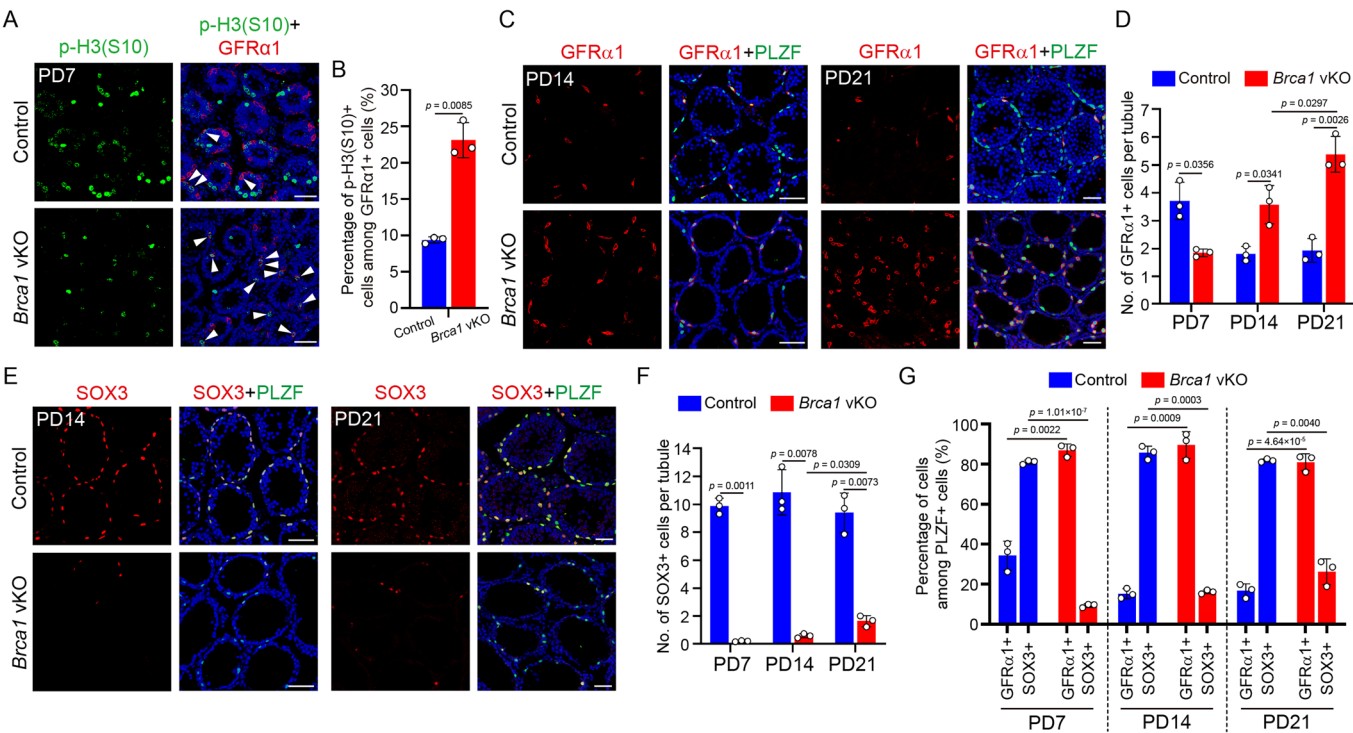

**Figure 4. Spermatogonial stem cells hyperproliferate after BRCA1 loss.**

(A) IF staining of the mitosis marker p-H3(S10) and GFRα1 in frozen sections of testes from Control and *Brca1* vKO male mice at PD7. The double-positive cells were marked by a white arrowhead. (B) Statistical analysis of the percentage of p-H3(S10)+ cells among GFRα1+ cells in testes of Control and *Brca1* vKO male mice at PD7. At least 50 GFRα1-positive cells from each mouse were analyzed. (C) IF staining of GFRα1 and PLZF in frozen sections of testes from Control and *Brca1* vKO male mice at PD14 and PD21. (D) Statistical analysis of the number of GFRα1+ cells per seminiferous tubule in testes of Control and *Brca1* vKO male mice at PD7, PD14, and PD21. (E) IF staining of SOX3 and PLZF in frozen sections of testes from Control and *Brca1* vKO males at PD14 and PD21. (F) Statistical analysis of the number of SOX3+ cells per seminiferous tubule in testis of Control and *Brca1* vKO male mice at PD7, PD14, and PD21. (G) Statistical analysis of the percentage of GFRα1+ and SOX3+ cells among PLZF+ cells in testes of Control and *Brca1* vKO male mice at PD7, PD14, and PD21. Data information: Data are presented as mean ± SD. Three mice of each genotype were analyzed. *P* value, two-tailed unpaired Student's *t* test. Scale bar, 50 μm. Source data are available online for this figure.

spermatogonia remained high in *Brca1* vKO testes (Fig. 7D–I), indicating that DNA damage repair was significantly delayed, consistent with DNA damage repair deficiency caused by BRCA1 loss. Importantly, the intensities of γH2AX signals and the percentages of γH2AX-positive cells were similar between both populations in *Brca1* vKO testes (Fig. 7E,F,H,I), indicating that DNA damage repair was similarly impaired in both populations in *Brca1* vKO testes.

Interestingly, compared with GFRα1-positive spermatogonial stem cells, SOX3-positive progenitor spermatogonia had a much higher percentage of p-H3(S10)-positive mitotic cells at PD7 (Fig. 7J,K), suggesting that these cells have a much higher proliferation rate. Since cell proliferation in the presence of unrepaired DNA damage is detrimental, the higher proliferation rate may lead to elevated cell death in progenitor spermatogonia when spontaneous DNA damage fails to be repaired efficiently in the absence of BRCA1.

## Formaldehyde is a potential source of spontaneous DNA damage during the formation of undifferentiated spermatogonia

The observations in *Brca1* vKO testes suggest that BRCA1-dependent DNA damage repair is essential for the formation of

undifferentiated spermatogonia. Since BRCA1 participates in multiple DNA damage repair pathways for various forms of DNA damage, the source of spontaneous DNA damage during the formation of undifferentiated spermatogonia is unclear. DNA replication stress is one of the primary sources of spontaneous DNA damage that is closely monitored and repaired. To examine if spontaneous DNA damage arises during DNA replication, we labeled the replicating cells with BrdU in the testes of WT male mice. Interestingly, while γH2AX signals are present in 80% of PLZF-positive undifferentiated spermatogonia at PD4 and PD7, only 30% of undifferentiated spermatogonia are positive for BrdU (Fig. 8A,B), suggesting that DNA damage is present in most undifferentiated spermatogonia regardless of their replication status. In addition, less than 30% of all BrdU-positive cells are PLZF-positive undifferentiated spermatogonia at PD4 and PD7 (Fig. 8A,B). The remaining BrdU-positive cells are Sertoli cells (Appendix Fig. S4A,B), in which no spontaneous DNA damage arises. Taking these observations together, DNA replication is unlikely to be the primary source of spontaneous DNA damage during the formation of undifferentiated spermatogonia.

Besides DNA replication, reactive metabolites can also attack DNA and generate DNA damage. Reactive oxygen species (ROS) are the most well-characterized reactive metabolites produced

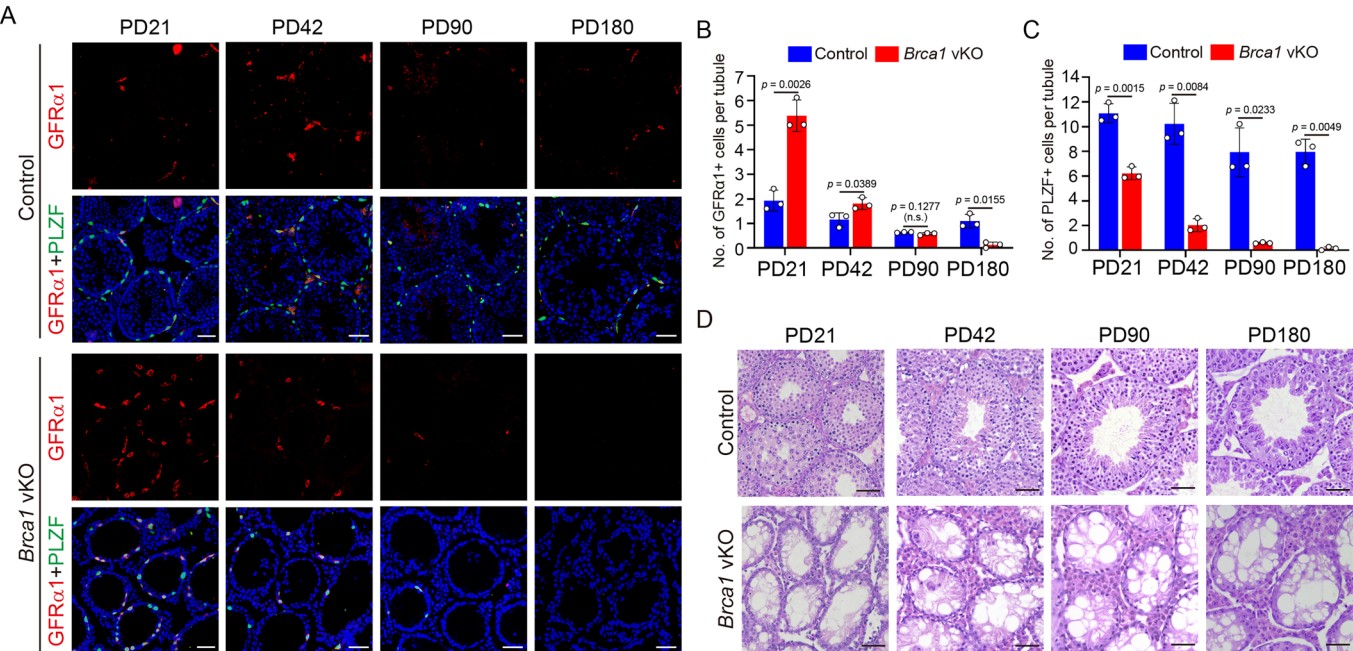

**Figure 5. Gradual loss of spermatogonial stem cells leads to premature exhaustion of undifferentiated spermatogonia in *Brca1* vKO testes.**

(A) IF staining of GFRα1 and PLZF in frozen sections of testes from Control and *Brca1* vKO male mice at PD21, PD42, PD90, and PD180. (B, C) Statistical analysis of the number of GFRα1+ cells (B) and PLZF+ cells (C) per seminiferous tubule in testis of Control and *Brca1* vKO male mice at PD21, PD42, PD90, and PD180. The GFRα1 data in control and *Brca1* vKO at PD21 were the same as in Fig. 4D. (D) H&E staining in paraffin sections of testes from Control and *Brca1* vKO male mice at PD21, PD42, PD90, and PD180. Data information: Data are presented as mean ± SD. Three mice of each genotype were analyzed. *P* value, two-tailed unpaired Student's *t* test. n.s. not significant. Scale bar, 50 μm. Source data are available online for this figure.

throughout the cell cycle (Schieber and Chandel, 2014). To examine if ROS are responsible for DNA damage, we tested if ROS scavenger N-Acetylcysteine (NAC) could alleviate the DNA damage during the formation of undifferentiated spermatogonia. Due to breastfeeding, we could not deliver NAC effectively to newborn and infant mice during the formation of undifferentiated spermatogonia in WT testes before PD14. Since DNA damage should continue to be generated in *Brca1* vKO testes at PD21 when undifferentiated spermatogonia was not formed, we examined if NAC treatment at this stage could alleviate the DNA damage and rescue the formation of undifferentiated spermatogonia in *Brca1* vKO testes. However, three weeks of NAC treatment did not increase the PLZF-positive undifferentiated spermatogonia and GFRα1-positive spermatogonial stem cells in *Brca1* vKO testes (Fig. 8C,D). Therefore, ROS are unlikely to be responsible for spontaneous DNA damage during the formation of undifferentiated spermatogonia.

Formaldehyde, a product of folate decomposition and protein demethylation, generates DNA damage independent of the cell cycle (Burgos-Barragan et al, 2017; Tibbetts and Appling, 2010). During the transition from gonocytes to undifferentiated spermatogonia and during the formation of undifferentiated spermatogonia, there should be a global change of transcription, which might be associated with massive histone demethylation and formaldehyde production. Therefore, we examined if formaldehyde was a source of spontaneous DNA damage during the formation of

undifferentiated spermatogonia that required BRCA1-dependent DNA damage repair. Since it is challenging to visualize DNA damage generated by formaldehyde directly, we took a similar approach as in previous studies to examine if removing ALDH2, an enzyme critical for formaldehyde clearance (Dingler et al, 2020), exacerbated the defects in *Brca1* vKO testes. Unlike *Brca1* vKO testes, *Aldh2* KO testes did not exhibit defects in undifferentiated spermatogonia at PD21 (Fig. 8E,F; Appendix Fig. S5A,B). However, much fewer PLZF-positive undifferentiated spermatogonia and GFRα1-positive spermatogonial stem cells were observed in *Brca1* vKO-*Aldh2* KO testes than in *Brca1* vKO testes at PD21 (Fig. 8E,F; Appendix Fig. S5A,B). Therefore, *Aldh2* KO exacerbated the defects in the formation of undifferentiated spermatogonia in *Brca1* vKO testes.

Although we predicted that more DNA damage is generated in the undifferentiated spermatogonia of *Brca1* vKO-*Aldh2* KO testes, there was no significant difference in the intensities of γH2AX signals and the percentages of γH2AX-positive undifferentiated spermatogonia between *Brca1* vKO and *Brca1* vKO-*Aldh2* KO testes (Appendix Fig. S5C,D). Since strong γH2AX signals are already present in *Brca1* vKO testes, the potential increase in DNA damage in *Brca1* vKO-*Aldh2* KO might not be easily detected by immunofluorescent staining, a semi-quantitative method. Collectively, these observations suggest that formaldehyde is a potential source of spontaneous DNA damage during the formation of undifferentiated spermatogonia. However, further investigation is required to validate this hypothesis.

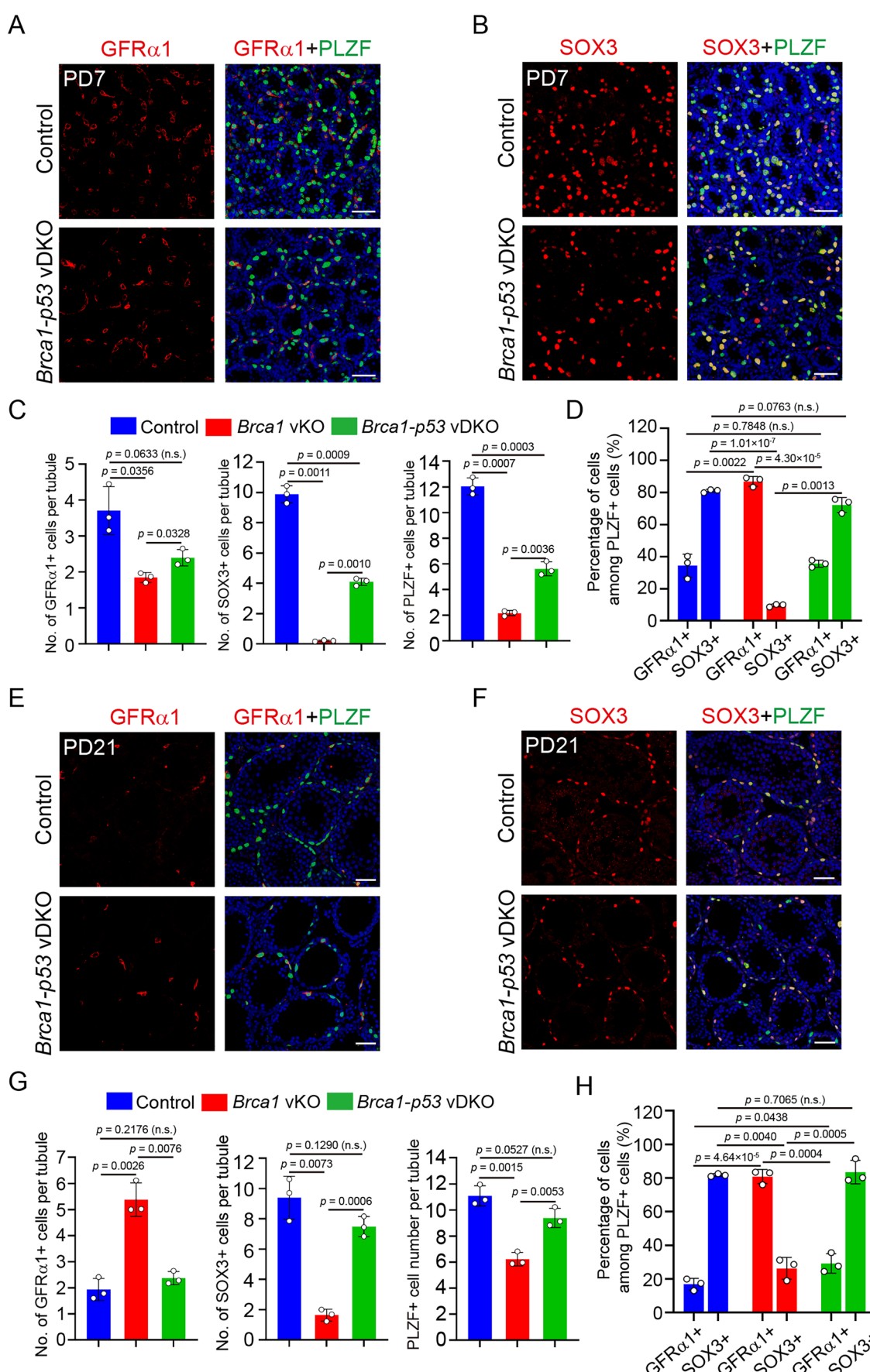

**Figure 6.    Blocking apoptotic cell death of progenitor spermatogonia restores the homeostasis of undifferentiated spermatogonia in *Brca1* vKO testes.**

(A) IF staining of GFRα1 and PLZF in frozen sections of testes from Control and *Brca1-p53* vDKO male mice at PD7. (B) IF staining of SOX3 and PLZF in frozen sections of testes from Control and *Brca1-p53* vDKO male mice at PD7. (C) Statistical analysis of the number of GFRα1 + , SOX3 + , and PLZF+ cells per seminiferous tubule in cross-sections of testes from Control, *Brca1* vKO, and *Brca1-p53* vDKO male mice at PD7. The GFRα1 data in control and *Brca1* vKO at PD7 were the same as in Figs. 3E and 4D. The SOX3 data in control and *Brca1* vKO at PD7 were the same as in Figs. 3E and 4F. The PLZF data in control and *Brca1* vKO at PD7 were the same as in Fig. 3D. (D) Statistical analysis of the percentage of GFRα1+ and SOX3+ cells among PLZF+ cells in testes from Control, *Brca1* vKO, and *Brca1-p53* vDKO male mice at PD7. The control and *Brca1* vKO data were the same as in Figs. 3F and 4G. (E) IF staining of GFRα1 and PLZF in frozen sections of testes from Control and *Brca1-p53* vDKO male mice at PD21. (F) IF staining of SOX3 and PLZF in frozen sections of testes from Control and *Brca1-p53* vDKO male mice at PD21. (G) Statistical analysis of the number of GFRα1+ and SOX3+ cells per seminiferous tubule in testes from Control, *Brca1* vKO, and *Brca1-p53* vDKO male mice at PD21. The GFRα1 data in control and *Brca1* vKO at PD21 were the same as in Figs. 4D and 5B. The SOX3 data in control and *Brca1* vKO at PD21 were the same as in Fig. 4F. The PLZF data in control and *Brca1* vKO at PD21 were the same as in Fig. 5C. (H) Statistical analysis of the percentage of GFRα1+ and SOX3+ cells among PLZF+ cells in testes from Control, *Brca1* vKO, and *Brca1-p53* vDKO male mice at PD21. The control and *Brca1* vKO data were the same as in Fig. 4G. Data information: Data are presented as mean ± SD. Three mice of each genotype were analyzed. *P* value, two-tailed unpaired Student's *t* test. n.s. not significant. Scale bar, 50 μm. Source data are available online for this figure.

## Discussion

### BRCA1-dependent repair of spontaneous DNA damage is essential for the formation of undifferentiated spermatogonia

As the source of uninterrupted spermatogenesis, undifferentiated spermatogonia are formed soon after birth. Since the integrity of the genome to be passed to the offspring needs to be protected against DNA damage that arises during the life span of undifferentiated spermatogonia, it is not surprising that mice deficient for several DNA damage repair proteins manifest defects in the maintenance of undifferentiated spermatogonia (Takubo et al, 2008; Tang et al, 2024). Paradoxically, our study reveals that massive spontaneous DNA damage arises during the formation of undifferentiated spermatogonia in mice, which poses a significant threat to genome integrity. Although no cell apoptosis is associated with DNA damage generation in WT mice, loss of BRCA1 compromises DNA damage repair and disrupts the formation of undifferentiated spermatogonia (Fig. 9), highlighting the importance of BRCA1-dependent DNA damage repair in protecting genome integrity during this process.

BRCA1 is the only DNA damage repair protein known to be required for the formation of undifferentiated spermatogonia. Given the multifaceted role of BRCA1 in DNA damage repair (Schreuder et al, 2024; Tarsounas and Sung, 2020), the requirement for BRCA1 during this process is likely determined by the nature of DNA damage. Our study suggests that formaldehyde produced from transcription change-associated histone demethylation is a potential source of spontaneous DNA damage during the formation of undifferentiated spermatogonia. With the change in cell identity, massive transcription reprogramming occurs during the formation of undifferentiated spermatogonia (Hammoud et al, 2014; Law et al, 2019). For example, PLZF is not expressed in gonocytes at E18.5 but starts to express in developing spermatogonia at PD1. When spermatogonial stem cells gradually transit to progenitor spermatogonia, GFRα1 expression is turned off, and SOX3 expression is turned on. We could not directly analyze the formaldehyde level in vivo due to technical limitations, but we speculate that massive transcription reprogramming during the formation of undifferentiated spermatogonia leads to a surge in formaldehyde level. Formaldehyde-induced DNA damage is generated during primordial germ cell (PGC) development and hematopoietic differentiation when dramatic changes in cell identities and massive transcription reprogramming occur (Hill and Crossan, 2019; Shen et al, 2020). It is possible that the formation of undifferentiated

spermatogonia is another developmental stage during which genome integrity needs to be protected against the inevitable formaldehyde-induced DNA damage. Further investigation is required to validate this hypothesis. It should be noted that besides histone demethylation, drastic epigenetic changes could occur during the formation of undifferentiated spermatogonia, which trigger global chromatin conformation changes and contribute in part to the pan-nuclear γH2AX signals observed.

Our observation is consistent with previous findings that BRCA1 is required for repairing aldehyde-induced DNA damage (Tacconi et al, 2017). Formaldehyde generates DNA interstrand crosslinks (ICLs) and DNA-protein crosslinks (DPCs). As a major pathway for ICL repair, the Fanconi anemia (FA) pathway functions in both PGC development and hematopoietic differentiation (Hill and Crossan, 2019; Shen et al, 2020), but its role during the formation of undifferentiated spermatogonia is not clear. Recent studies suggest that formaldehyde-induced DPCs are repaired by transcription-coupled repair (Carnie et al, 2024; Oka et al, 2024; van Sluis et al, 2024). BRCA1 belongs to the FA protein family and functions in ICL repair (Che et al, 2018; Semlow and Walter, 2021). BRCA1 has also been linked with transcription-coupled repair in previous studies (Wei et al, 2011). It will be interesting to examine if BRCA1 functions in the FA pathway and transcription-coupled repair during the formation of undifferentiated spermatogonia.

### BRCA1 loss differentially impacts spermatogonial stem cells and progenitor spermatogonia

During the formation of undifferentiated spermatogonia, spermatogonial stem cells gradually transit into progenitor spermatogonia until homeostasis is established between these two populations. Due to the dramatic reduction of progenitor spermatogonia after BRCA1 loss, the homeostasis of undifferentiated spermatogonia failed to be established. Our study suggests that the high proliferation rate potentially contributes to the susceptibility of the progenitor spermatogonia to BRCA1 loss. DNA damage usually activates cell cycle checkpoints to arrest cell proliferation (Sperka et al, 2012), but no cell proliferation arrest is observed during the formation of undifferentiated spermatogonia despite the presence of massive DNA damage, suggesting that DNA damage is repaired promptly during this process. Given the dual functions of BRCA1 in DNA damage repair and cell cycle checkpoint activation (Huen et al, 2010; Savage and Harkin, 2015), it is likely that *Brca1* KO cells continue to proliferate in the presence of unrepaired DNA damage and eventually die. Mutations in the coiled-coil domain of

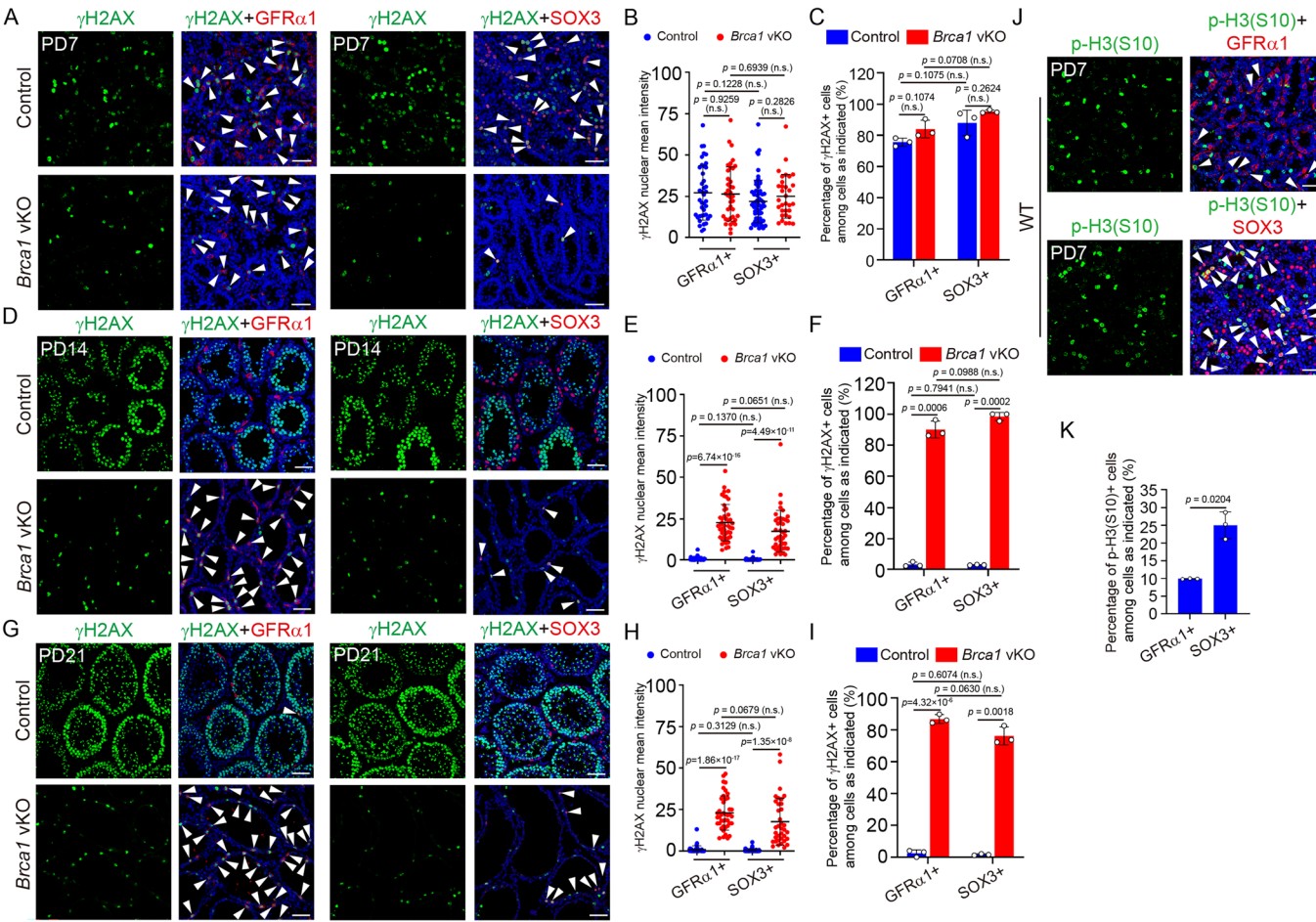

**Figure 7. The rapid proliferation of progenitor spermatogonia potentially contributes to their susceptibility to BRCA1 loss.**

(A) Co-staining of GFRα1&γH2AX and SOX3&γH2AX in frozen sections of testes from Control and *Brca1* vKO male mice at PD7. The double-positive cells were marked by a white arrowhead. (B) Statistical analysis of γH2AX nuclear mean intensity of GFRα1+ and SOX3+ cells in testes from Control and *Brca1* vKO male mice at PD7. At least 30 positive cells from each genotype were analyzed. (C) Statistical analysis of the percentage of γH2AX+ cells among GFRα1+ and SOX3+ cells in testes from Control and *Brca1* vKO male mice at PD7. Three mice of each genotype were analyzed. (D) Co-staining of GFRα1&γH2AX and SOX3&γH2AX in frozen sections of testes from Control and *Brca1* vKO male mice at PD14. The double-positive cells were marked by a white arrowhead. (E) Statistical analysis of γH2AX nuclear mean intensity of GFRα1+ cells and SOX3+ cells in testes from Control and *Brca1* vKO male mice at PD14. At least 30 positive cells from each genotype were analyzed. (F) Statistical analysis of the percentage of γH2AX+ cells among GFRα1+ and SOX3+ cells in testes from Control and *Brca1* vKO male mice at PD14. Three mice of each genotype were analyzed. (G) Co-staining of GFRα1&γH2AX and SOX3&γH2AX in frozen sections of testes from Control and *Brca1* vKO male mice at PD21. The double-positive cells were marked by a white arrowhead. (H) Statistical analysis of γH2AX nuclear mean intensity of GFRα1+ cells and SOX3+ cells in testis sections of Control and *Brca1* vKO male mice at PD21. At least 30 positive cells from each genotype were analyzed. (I) Statistical analysis of the percentage of γH2AX+ cells among GFRα1+ and SOX3+ cells in testes from Control and *Brca1* vKO male mice at PD21. Three mice of each genotype were analyzed. (J) Co-staining of GFRα1& p-H3(S10) and SOX3&p-H3(S10) in frozen sections of testes from WT male mice at PD7. The double-positive cells were marked by a white arrowhead. (K) Statistical analysis of the percentage of p-H3(S10)+ cells among GFRα1+ and SOX3+ cells in testes from WT male mice at PD7. Three mice were analyzed. Data information: Data are presented as mean ± SD. *P* value, two-tailed unpaired Student's *t* test. n.s. not significant. Scale bar, 50 μm. Source data are available online for this figure.

BRCA1, such as L1363P of mouse BRCA1, disrupt BRCA1's function in DNA damage repair but not cell cycle checkpoint (Park et al, 2020; Pulver et al, 2021). It will be interesting to examine the impact on the formation of undifferentiated spermatogonia in *Brca1*^L1363P/L1363P mice.

Unlike the dramatic reduction of progenitor spermatogonia after BRCA1 loss, spermatogonial stem cells are mildly reduced. In addition, spermatogonial stem cells hyperproliferate, which could be interpreted as an attempt to compensate for the dramatic reduction of progenitor spermatogonia. Spermatogonial stem cells are gradually lost afterward, and undifferentiated spermatogonia are exhausted prematurely. Although it has been shown that

hyperproliferation leads to aging and exhaustion of hematopoietic stem cells (Singh et al, 2018), it has never been demonstrated in spermatogonial stem cells. It should be noted that unrepaired DNA damage is also present in spermatogonial stem cells in *Brca1* vKO testes, which may also contribute to the premature exhaustion of undifferentiated spermatogonia. Therefore, we could not conclude that hyperproliferation of spermatogonial stem cells alone leads to premature exhaustion of undifferentiated spermatogonia in *Brca1* vKO testes. It will be interesting to deplete progenitor spermatogonia using genetic mouse models without causing DNA damage repair defects and test if dramatic reduction of progenitor

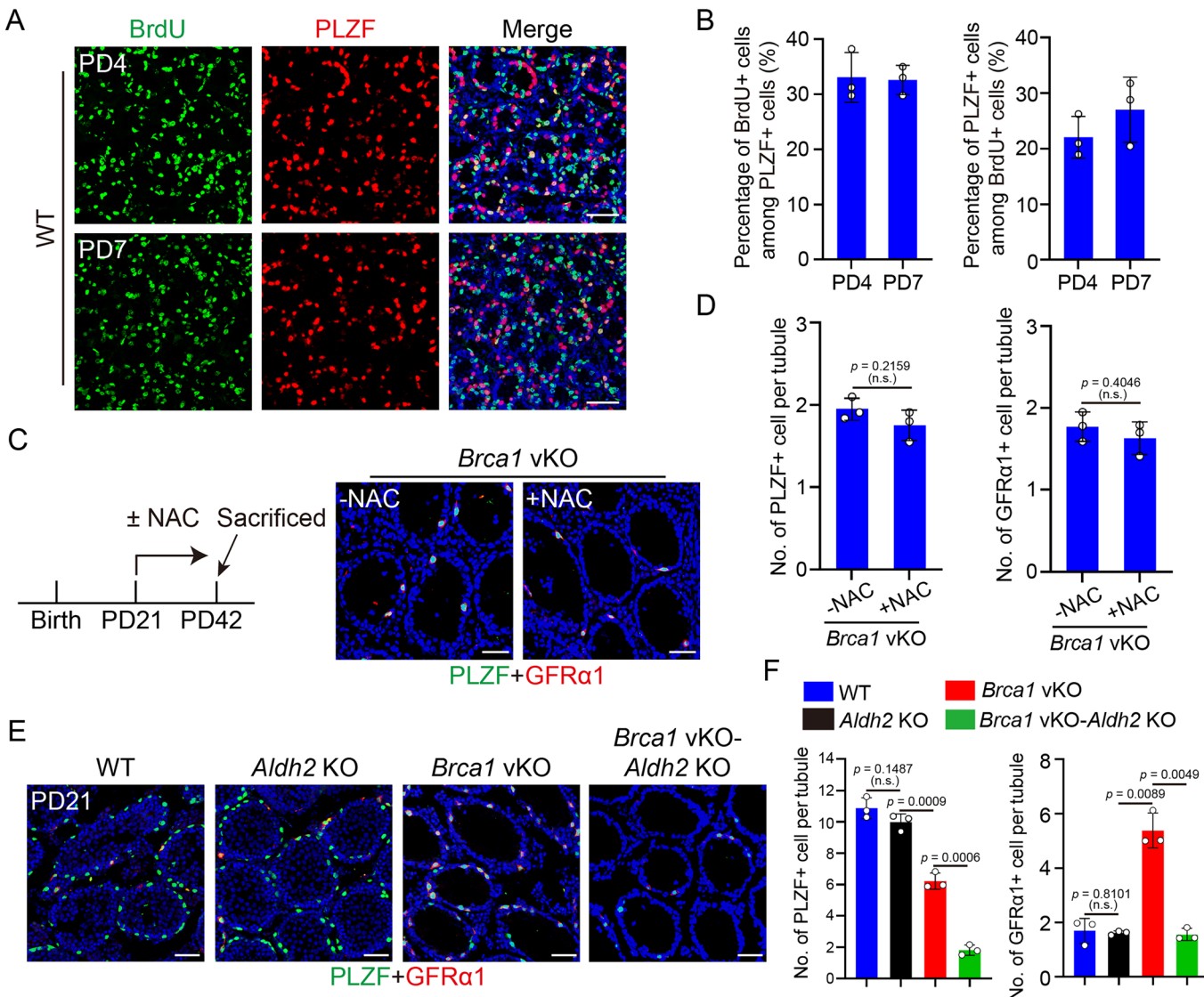

**Figure 8. Aldehyde is likely a source of spontaneous DNA damage during the formation of undifferentiated spermatogonia.**

(A) IF staining of BrdU and PLZF in frozen sections of testes from WT mice at PD4 and PD7. (B) Statistical analysis of the percentage of BrdU+ cells among PLZF+ cells and the percentage of PLZF+ cells among BrdU+ cells in testes from WT mice at PD4 and PD7. 3 mice were analyzed. (C) (left panel) Strategy to feed the *Brca1* vKO male mice with drinking water containing NAC. (right panel) IF staining of PLZF&GFRα1 in frozen sections of testes from male mice fed with or without NAC for three weeks. (D) Statistical analysis of the number of PLZF+ cells and GFRα1+ cells per seminiferous tubule in cross-sections of testes from *Brca1* vKO male mice fed with or without NAC. 3 mice from each group were analyzed. (E, F) IF staining of PLZF&GFRα1 in frozen sections of testes from WT, *Aldh2* KO, *Brca1* vKO, and *Brca1* vKO-*Aldh2* KO male mice at PD21 and statistical analysis of the number of PLZF+ cells and GFRα1+ cells per seminiferous tubule in cross-sections of testes. The *Brca1* vKO data were the same as in Figs. 4D and 5B,C. Three mice of each genotype were analyzed. Data information: Data are presented as mean ± SD. *P* value, two-tailed unpaired Student's *t* test. n.s. not significant. Scale bar, 50 μm. Source data are available online for this figure.

spermatogonia triggers hyperproliferation of spermatogonial stem cells and leads to premature exhaustion of undifferentiated spermatogonia.

## Exogenous DNA damage may generate differential impacts on spermatogonial stem cells and progenitor spermatogonia

Although undifferentiated spermatogonia repair spontaneous DNA damage promptly during their formation, they are extremely sensitive to DNA damage after formation. DNA alkylating reagent busulfan, commonly used to prepare recipients for spermatogonial stem cell transplantation in mice, effectively depletes most undifferentiated spermatogonia without causing global toxicity (La and Hobbs, 2019; Sriram et al, 2024). However, the DNA damage response remains to be characterized in undifferentiated spermatogonia after they are formed. Since spermatogonial stem cells and progenitor spermatogonia can be easily distinguished in mice, it will be important to examine if progenitor spermatogonia are more susceptible to DNA damage than spermatogonial stem

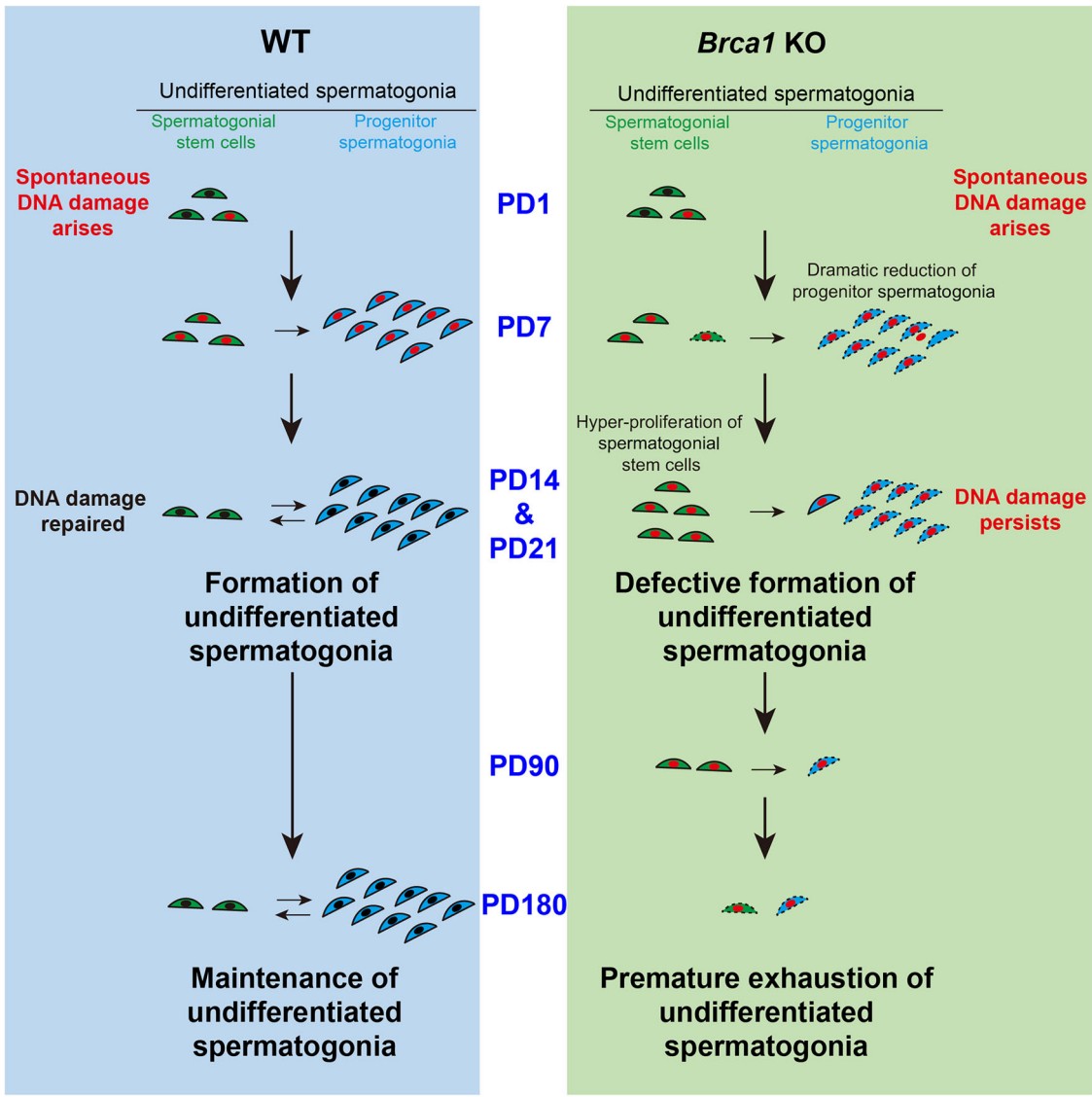

**Figure 9. Working model: BRCA1 is indispensable for the formation of undifferentiated spermatogonia.**

During the formation of undifferentiated spermatogonia in WT mice, spermatogonial stem cells gradually transit into progenitor spermatogonia until homeostasis is established between these two populations. Massive spontaneous DNA damage arises during this process (PD1-PD7), but it is repaired promptly before the formation of undifferentiated spermatogonia (PD14-PD21). Undifferentiated spermatogonia are then maintained to support uninterrupted spermatogenesis (PD90-PD180). Brca1 KO compromises DNA damage repair so that DNA damage persists (PD14-PD21). The dramatic reduction of progenitor spermatogonia disrupts the homeostasis between the two populations. Despite the hyperproliferation of spermatogonial stem cells, undifferentiated spermatogonia fail to form (PD14-PD21). Spermatogonial stem cells are eventually depleted, leading to premature exhaustion of undifferentiated spermatogonia (PD90-PD180).

cells after these two populations have reached homeostasis. After acute exposure to DNA damage that does not deplete all undifferentiated spermatogonia, male fertility drops but gradually restores (La and Hobbs, 2019). It remains to be examined if spermatogonial stem cells hyperproliferate in this scenario and if the hyperproliferation of spermatogonial stem cells leads to premature exhaustion of undifferentiated spermatogonia.

**BRCA1 loss is compatible with cell proliferation in vivo**

Besides the formation of undifferentiated spermatogonia, BRCA1 is important for multiple developmental processes (Pao et al, 2014;

Shukla et al, 2011; Sotiropoulou et al, 2013; Vasanthakumar et al, 2016). In particular, BRCA1 loss leads to embryonic lethality (Hakem et al, 1996; Liu and Lu, 2020; Ludwig et al, 1997), but the reasons are unclear. Since BRCA1 loss is incompatible with cell proliferation in vitro, it is generally believed that BRCA1 loss disrupts some essential cellular activities and leads to cell proliferation defects during embryonic development (Hakem et al, 1996). However, *Brca1* KO embryos die at E6.5, before which cell proliferation should be normal. In this study, we show that spermatogonial stem cells in *Brca1* KO testes are not eliminated until PD180, and these cells proliferate even faster than those in control testes at PD7. Therefore, BRCA1 loss is compatible with cell

proliferation in vivo, suggesting that the lethality of *Brca1* KO embryos is unlikely due to cell proliferation defects alone. Instead, we show that BRCA1 is required to repair spontaneous DNA damage during the formation of undifferentiated spermatogonia, which is potentially generated by formaldehyde produced from transcription change-associated histone demethylation. Since the growth of embryos is accompanied by dramatic changes in cell identity and massive transcription reprogramming, it will be interesting to examine if formaldehyde-induced spontaneous DNA damage is generated that requires BRCA1-dependent DNA damage repair during early embryonic development.

# Methods

## Reagents and tools table

| Reagent/resource | Reference or source | Identifier or catalog number |
|---|---|---|
| **Experimental models** | | |
| *Brca1*<sup>flox/flox</sup> mice | NCI Mouse Repository | 01XB8 |
| *Vasa-Cre* transgenic mice | The Jackson Laboratory, | 006954 |
| *p53*<sup>flox/flox</sup> mice | NCI Mouse Repository | 01XC2 |
| *H2ax*<sup>−/−</sup> mice | Celeste et al, 2002 | N/A |
| *Aldh2*<sup>−/−</sup> mice | Kitagawa et al, 2000 | N/A |
| **Antibodies** | | |
| Rabbit anti-γH2AX | Abcam | ab11174 |
| Rabbit anti-MVH | Abcam | ab13840 |
| Rat anti-GCNA | Abcam | ab82527 |
| Rabbit anti-BRCA1 | Wu et al, 2009 | N/A |
| Rabbit anti-PLZF | Santa Cruz | sc22839 |
| Goat anti-PLZF | R&D Systems | AF2944 |
| Goat anti-KIT | R&D Systems | AF1356 |
| Goat anti-GFRα1 | R&D Systems | AF560 |
| Goat anti-SOX3 | R&D Systems | AF2569 |
| Rabbit anti-cleaved PARP1 (Asp214) | Cell Signaling Technology, | 9544 |
| Rabbit anti-phospho-Histone H3 (Ser10) | Cell Signaling Technology | 9701S |
| Goat anti-SOX9 | R&D Systems | AF3075 |
| Rabbit anti-SOX9 | Sigma-Aldrich | AB5535 |
| Rat anti-BrdU | Abcam | ab6326 |
| Alexa Fluor 594 Donkey Anti-Goat IgG (H + L) | Jackson Immunoresearch | 705-585-147 |
| Alexa Fluor 488 Donkey Anti-Rabbit IgG (H + L) | Jackson Immunoresearch | 711-545-152 |
| Alexa Fluor 594 Goat Anti-Rat IgG (H + L) | Jackson Immunoresearch | 112-585-167 |
| Alexa Fluor® 488 Goat Anti-Rat IgG H&L | Abcam | ab150157 |

| Reagent/resource | Reference or source | Identifier or catalog number |
|---|---|---|
| **Chemicals, enzymes, and other reagents** | | |
| N-acetylcysteine (NAC) | Sigma-Aldrich | A7250 |
| BrdU | Sigma-Aldrich | B5002 |
| Hoechst 33342 | Invitrogen | H21492 |
| FluorSave Reagent | Merck millipore | 345789 |
| Collagenase IV | Sangon Biotech | A004186 |
| DNase I | Invitrogen | AM2222 |
| TrypLE™ Express | Gibco | 12605028 |
| Bouin's Solution | Sigma | HT10132 |
| OCT compound | SAKURA | #4583 |
| **Software** | | |
| GraphPad Prism 9.5 | GraphPad software | N/A |
| Adobe Illustrator CS6 | Adobe | N/A |

## Mice

All mice were housed under an appropriately controlled environment (temperature 20–22 °C, humidity 50–70%, and 12 h light/12 h dark cycle) with easy access to food and water. All experimental procedures were approved by the Zhejiang University Animal Care and Use Committee (File No. ZJU20190144). Littermates were used as controls.

## Histological analysis

Testes or epididymides of male mice with indicated ages were isolated, fixed in Bouin's Solution for 48 h at room temperature, dehydrated stepwise through an ethanol series (30, 50, 70, 90%, and 100% ethanol), embedded in paraffin, and sectioned. After dewaxing and hydration, the sections were stained with hematoxylin and 1% eosin and imaged with a Leica DM4000 upright microscope.

## BrdU incorporation assay

BrdU (Sigma) was injected intraperitoneally into male mice at 50 μg/g body weight, and testes were isolated from mice 4 h later.

## NAC feeding

Mice were fed with N-acetylcysteine (NAC) in drinking water at 1 mg/ml for three consecutive weeks before being sacrificed. Fresh NAC solution was prepared every other day.

## Immunofluorescence (IF) staining of frozen sections

The isolated testes were fixed in 4% paraformaldehyde (PFA) overnight at 4 °C, dehydrated in 30% sucrose, embedded in OCT compound, and sectioned. The sections were permeabilized in 0.5% Triton X-100 for 15 min, blocked with 5% BSA in PBS for 1 h at room temperature, and incubated overnight at room temperature with primary antibodies diluted in 1% BSA. Following three washes

with PBS, Alexa Fluor-labeled secondary antibodies were added to the sections and incubated for 2 h at room temperature. The sections were then washed in PBS three times, incubated in 5 μg/ml of Hoechst 33342 for 10 min at room temperature, and mounted with FluorSave Reagent. Images were taken using a laser scanning confocal microscope (STELLARIS 5, Leica Microsystems). For quantification in frozen sections, the positive cells in at least 20 intact seminiferous tubules were counted. The number of positive cells per tubule was calculated by dividing the total number of positive cells by the number of seminiferous tubules analyzed. The data obtained from three mice were presented. The γH2AX nuclear mean intensity was quantified by ImageJ. The quantification data were available in Dataset EV1.

### IF staining of germ cells after DNA damage

PD1 testes were digested in PBS containing 1 mg/ml collagenase IV and 0.5 mg/ml DNase I at 37 °C for 5 min. Individual tubules were collected by brief centrifugation and were attached to coverslips overnight in DMEM containing 10% FBS. The cells were treated with 10 Gy ionizing radiation, fixed 4 h later, and subjected to IF staining.

### Whole-mount IF staining of seminiferous tubules

Tunica albuginea was removed from the isolated testes, and untangled seminiferous tubules were fixed in 4% PFA at 4 °C for 4 h. The seminiferous tubules were washed in PBS three times, permeabilized in 0.5% Triton X-100, blocked in 5% BSA for 1 h at room temperature, and incubated with primary antibodies diluted in 1% BSA at room temperature overnight. After washing in PBS three times, the seminiferous tubules were incubated with Alexa Fluor-labeled secondary antibodies for 1 h at room temperature. Hoechst 33342 was used to stain DNA. The seminiferous tubules were washed in PBS, mounted, and observed using a laser scanning confocal microscope (STELLARIS 5, Leica Microsystems). For quantification in isolated seminiferous tubules, the positive cells were counted in seminiferous tubules that are at least 1 mm by length. The number of positive cells per 100 μm seminiferous tubule was calculated by dividing the total number of all positive cells by the length of the seminiferous tubules. The numbers obtained from three mice of each genotype were presented. The quantification data were available in Dataset EV1.

### Single-cell RNA sequencing (scRNA-seq)

Testes of PD7 control and *Brca1* vKO mice were isolated and digested into single cells by two continuous steps with collagenase IV and TrypLE™ Express. The single-cell suspensions were loaded onto the 10× Genomics Chromium platform to partition into Gel Bead-In-Emulsions (GEMs), followed by barcoded reverse transcription and PCR amplification of cDNA for library construction according to the manufacturer's instructions. To mitigate the potential issue of capturing an insufficient number of germ cells in *Brca1* vKO samples due to smaller testicular volumes, we increased the number of cells from *Brca1* vKO that was loaded onto the sequencer. The single-cell libraries were sequenced by an MGISEQ-2000RS sequencer (MGI Tech). The raw sequencing data were aligned to the mouse reference genome mm10 (10× Genomics pre-built, version refdata-gex-mm10-2020-A), and count matrices were generated using the 10× Genomics Cell Ranger Pipeline (version 6.0.2). Subsequently, these matrices were

imported into R (version 4.2.3) via RStudio (version 2023.03.0 Build 386). R Package Seurat (version 4.3.0.1) was utilized for downstream scRNA-seq analysis (Hao et al, 2021). Cells with fewer than 750 unique genes detected or more than 10% mitochondrial gene expression were deemed low-quality and thus excluded. Cell cycle scores were computed by the Seurat function CellCycleScoring. The two samples were merged, normalized, and scaled, and RunPCA function was performed to select the principal components. Batch effects were corrected by Harmony using the Seurat embedded function RunHarmony (Korsunsky et al, 2019). Dimension reduction analysis was performed using RunTSNE. Cell clustering was accomplished by applying FindNeighbors and FindClusters, with a resolution set to 0.25. Canonical gene markers were plotted to assign the identity of cell clusters. The germ cell population was identified based on the expression of *Ddx4* and *Dazl*. Subsequently, the germ cell population was isolated and subjected to further clustering analysis to identify potential sub-clusters. A similar method as described previously was applied, with the resolution set to 0.2 in the FindClusters function. Cluster visualization was conducted on a t-distributed stochastic neighbor embedding (tSNE) plot within Seurat. Differentially expressed (DE) gene analysis between control and *Brca1* vKO testes within each cell cluster was performed by the FindMarkers function. In each cluster, DE genes with a *P* value of less than 0.05 were selected for pathway enrichment analysis using the DAVID web server (Huang da et al, 2009; Sherman et al, 2022).

### Statistical analysis

Unless otherwise specified in the text, a two-tailed unpaired Student's *t* test was used to evaluate the statistical significance between two groups of continuous data with normal distribution. A chi-squared test was used to evaluate the difference in the distribution of germ cells between two groups in scRNA-seq. No blinding was used in the experiments.

# Data availability

The scRNA-seq data reported in this study has been deposited in the Gene Expression Omnibus website with accession code GSE273607.

The source data of this paper are collected in the following database record: biostudies:S-SCDT-10_1038-S44319-025-00487-5.

# Peer review information

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

## Acknowledgements

We thank Jinjin Zheng and Junbin Qian for helping with single-cell RNA sequencing. We thank Quanfeng Zhu and Shenghui Hong for helping with mouse strain rederivation. We thank Chao Bi and Xiaoli Hong from the Core Facilities, Zhejiang University School of Medicine for their technical support. We thank Xiaofen Jin, Yumiao Niu, and Feiting Xie from the Central Laboratory, Women's Hospital, Zhejiang University School of Medicine for their technical support. This work is funded by Zhejiang Provincial Natural Science Foundation (LQ21C120001) and National Natural Science Foundation of China (32070829, 32100676, and 82171596).

## Author contributions

**Peng Li**: Formal analysis; Investigation; Writing—original draft; Writing—review and editing. **Licun Song**: Formal analysis; Investigation. **Longfei Ma**: Investigation. **Chunsheng Han**: Formal analysis; Methodology. **Lejun Li**: Conceptualization; Formal analysis; Supervision. **Lin-Yu Lu**: Conceptualization; Formal analysis; Supervision; Writing—original draft; Writing—review and editing. **Yidan Liu**: Conceptualization; Formal analysis; Supervision; Writing—original draft; Writing—review and editing.

Source data underlying figure panels in this paper may have individual authorship assigned. Where available, figure panel/source data authorship is listed in the following database record: biostudies:S-SCDT-10_1038-S44319-025-00487-5.

## Disclosure and competing interests statement

The authors declare no competing interests.

# Expanded View Figures

**Figure EV1.** **Spermatogonial stem cells and progenitor spermatogonia achieve homeostasis when undifferentiated spermatogonia is formed after birth.** ▶

(A) Cell identity and marker gene expression before and after the formation of undifferentiated spermatogonia, which contain spermatogonial stem cells and progenitor spermatogonia that achieve homeostasis. (B) IF staining of PLZF and GCNA in frozen sections of testes from WT mice at E18.5 and PD1. (C) IF staining of GFRα1 and PLZF in frozen sections of testes from WT mice at PD1, PD4, PD7, PD14, and PD21. (D) IF staining of SOX3 and PLZF in frozen sections of testes from WT mice at PD1, PD4, PD7, PD14, and PD21. (E) Statistical analysis of the percentage of GFRα1+ and SOX3+ cells among PLZF+ cells in testes from WT mice at PD1, PD4, PD7, PD14, and PD21. Data information: Data are presented as mean ± SD. Three mice at each timepoint were analyzed. Scale bar, 50 μm. n.s., not significant.

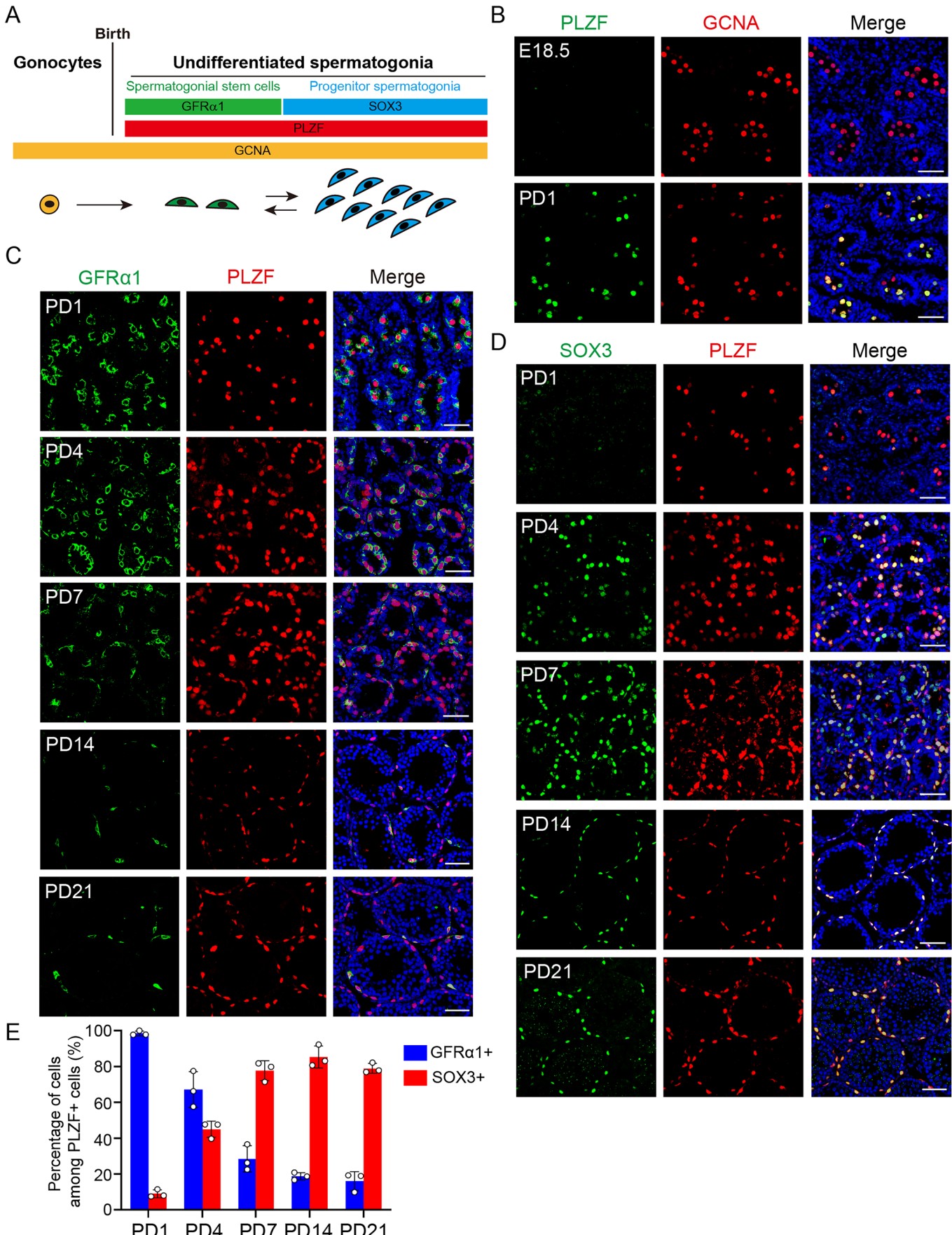

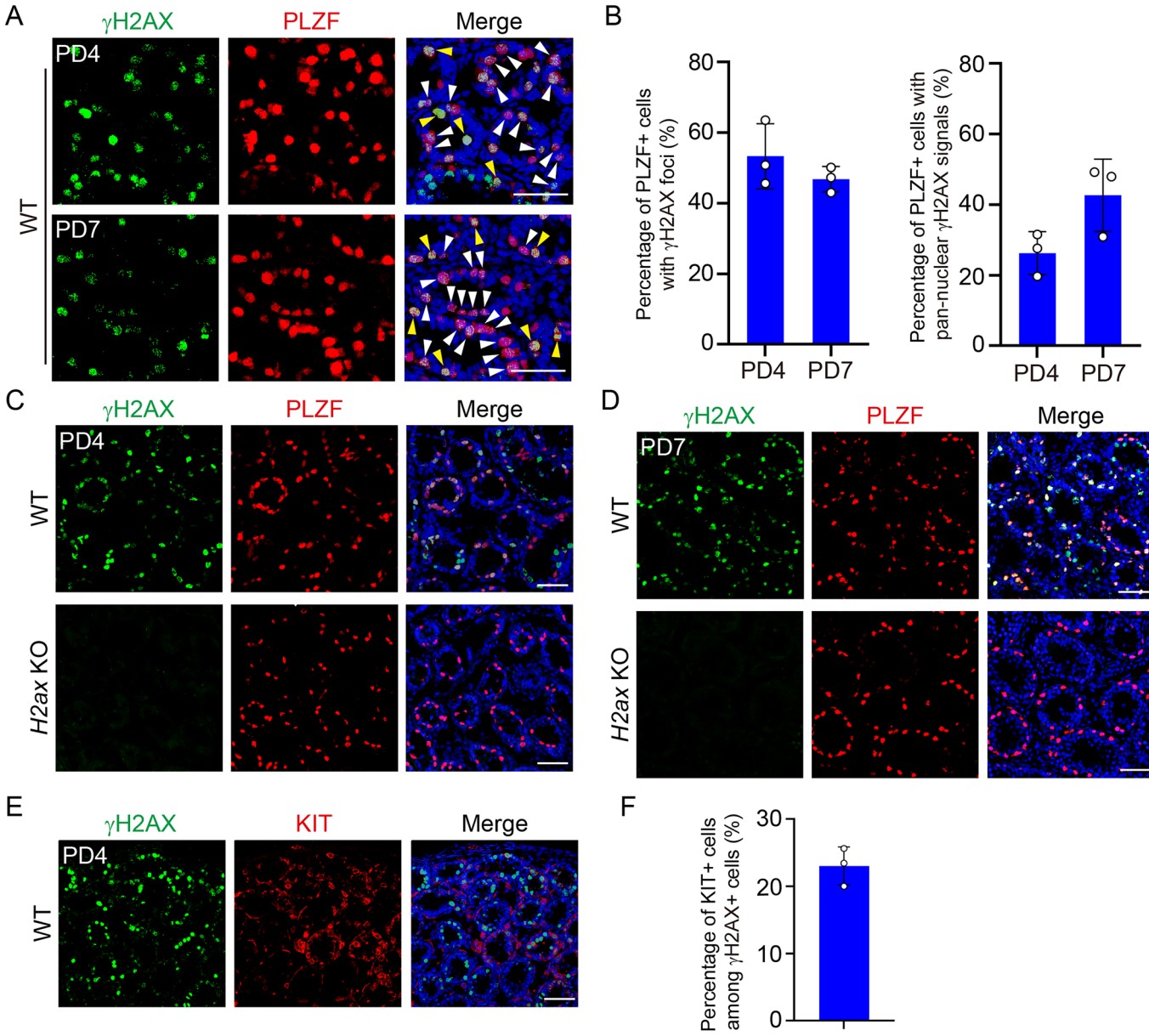

**Figure EV2. γH2AX signals are not observed during the formation of undifferentiated spermatogonia in *H2ax* KO mice.**

(**A**) Representative images of γH2AX and PLZF staining in frozen sections of testes from WT mice at PD4 and PD7. PLZF+ cells with γH2AX+ foci were labeled by white arrowhead and PLZF+ cells with pan-nuclear γH2AX+ signals were labeled by yellow arrowhead. (**B**) Statistical analysis of the percentage of PLZF+ cells with γH2AX+ foci or with pan-nuclear γH2AX+ signals in WT mice at PD4 and PD7. (**C, D**) IF staining of γH2AX and PLZF in frozen sections of testes from WT and *H2ax* KO mice at PD4 (**C**) and PD7 (**D**). (**E, F**) IF staining of γH2AX and KIT in frozen sections of testes from WT mice at PD4 and statistical analysis of the percentage of KIT+ cells among γH2AX+ cells. Data information: Data are presented as mean ± SD. 3 mice were analyzed. Scale bar, 50 μm.

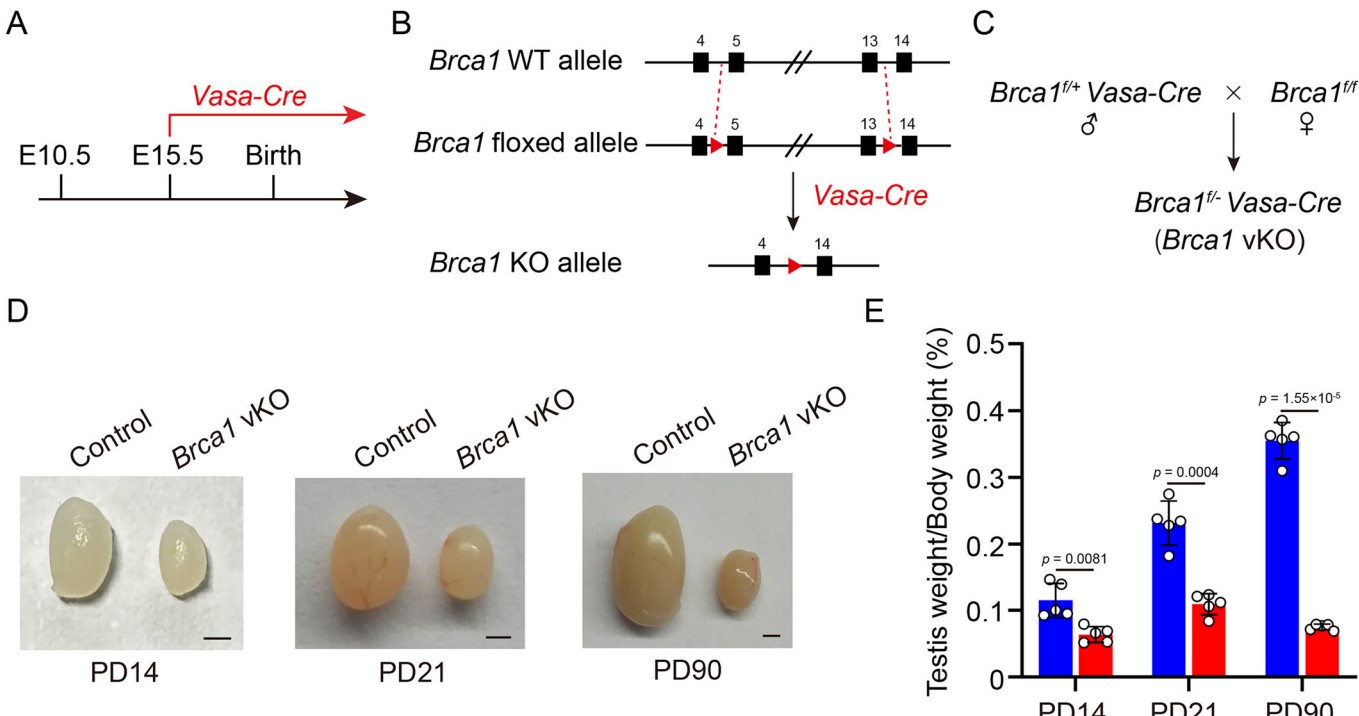

**Figure EV3. Generation of *Brca1* germ cell-specific KO mice.**

(A) Schematic diagram showing the expression timepoint of *Vasa-Cre* during mouse germ cell development. (B) Schematic illustrations of *Vasa-Cre* recombinase-mediated deletion of *Brca1* floxed allele. (C) Schematic illustrations of mating strategies to obtain *Brca1* vKO mice. (D) Representative images of testes from Control and *Brca1* vKO male mice at PD14, PD21, and PD90. Scale bar, 1 mm. (E) Statistical analysis of ratios of testis weight to body weight of Control and *Brca1* vKO male mice at PD14, PD21, and PD90. 5 mice of each genotype were analyzed. Data information: Data are presented as mean ± SD. *P* value, two-tailed unpaired Student's *t* test.

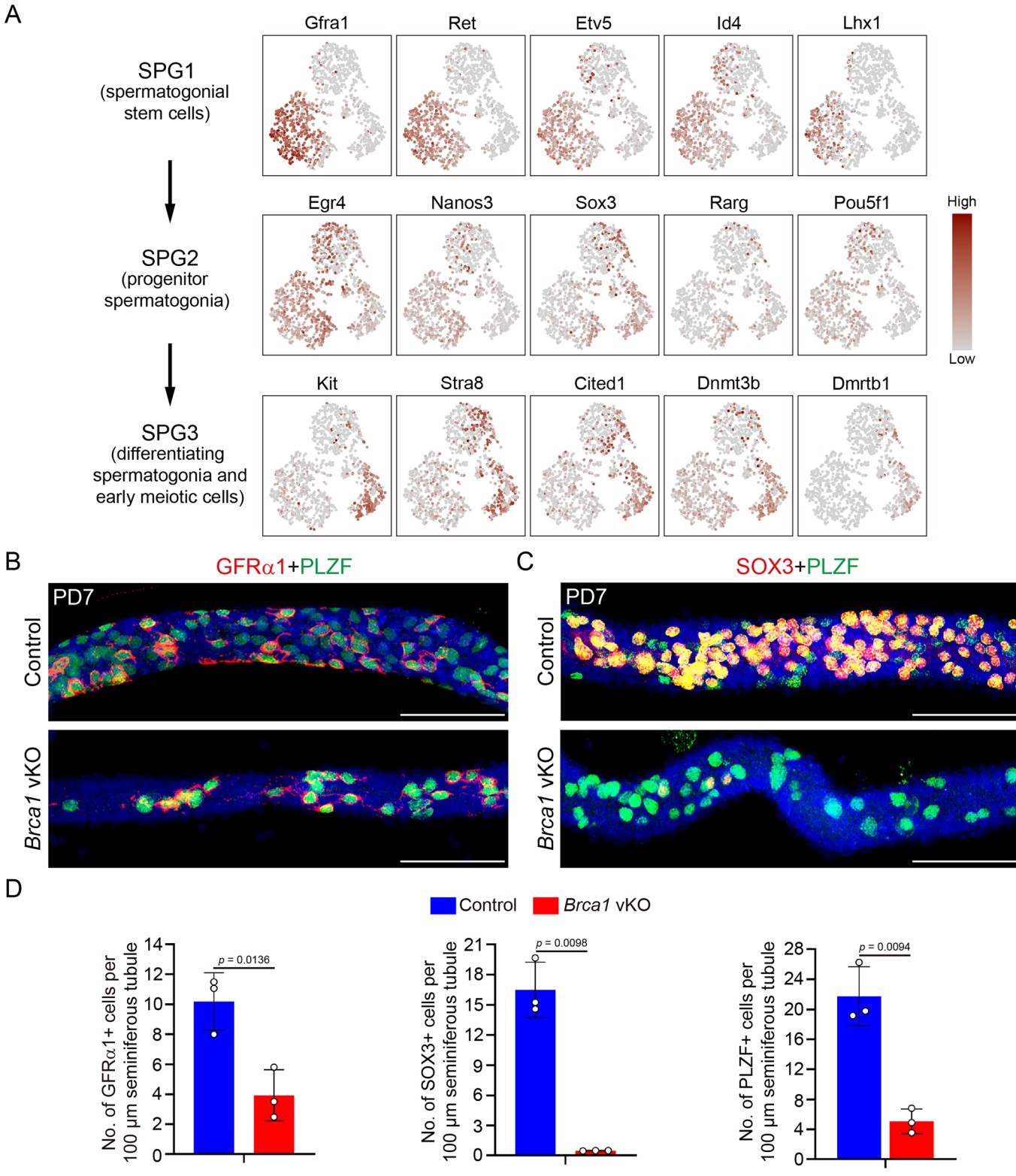

◄ **Figure EV4. Progenitor spermatogonia have a much more dramatic reduction than spermatogonial stem cells after BRCA1 loss.**

(A) Gene expression patterns of selected marker genes for each subtype of germ cells. The tSNE plot is in reference to all germ cells (control + *Brca1* vKO). (B) Whole-mount IF staining of GFRα1 and PLZF in seminiferous tubules of testes from Control and *Brca1* vKO male mice at PD7. (C) Whole-mount IF staining of SOX3 and PLZF in seminiferous tubules of testes from Control and *Brca1* vKO male mice at PD7. (D) Statistical analysis of the number of GFRα1 + , SOX3+cells, and PLZF+ cells per 100 μm seminiferous tubule in Control and *Brca1* vKO male mice at PD7. Three mice of each genotype were analyzed. Data information: Data are presented as mean ± SD. *P* value, two-tailed unpaired student's t test. Scale bar, 100 μm.

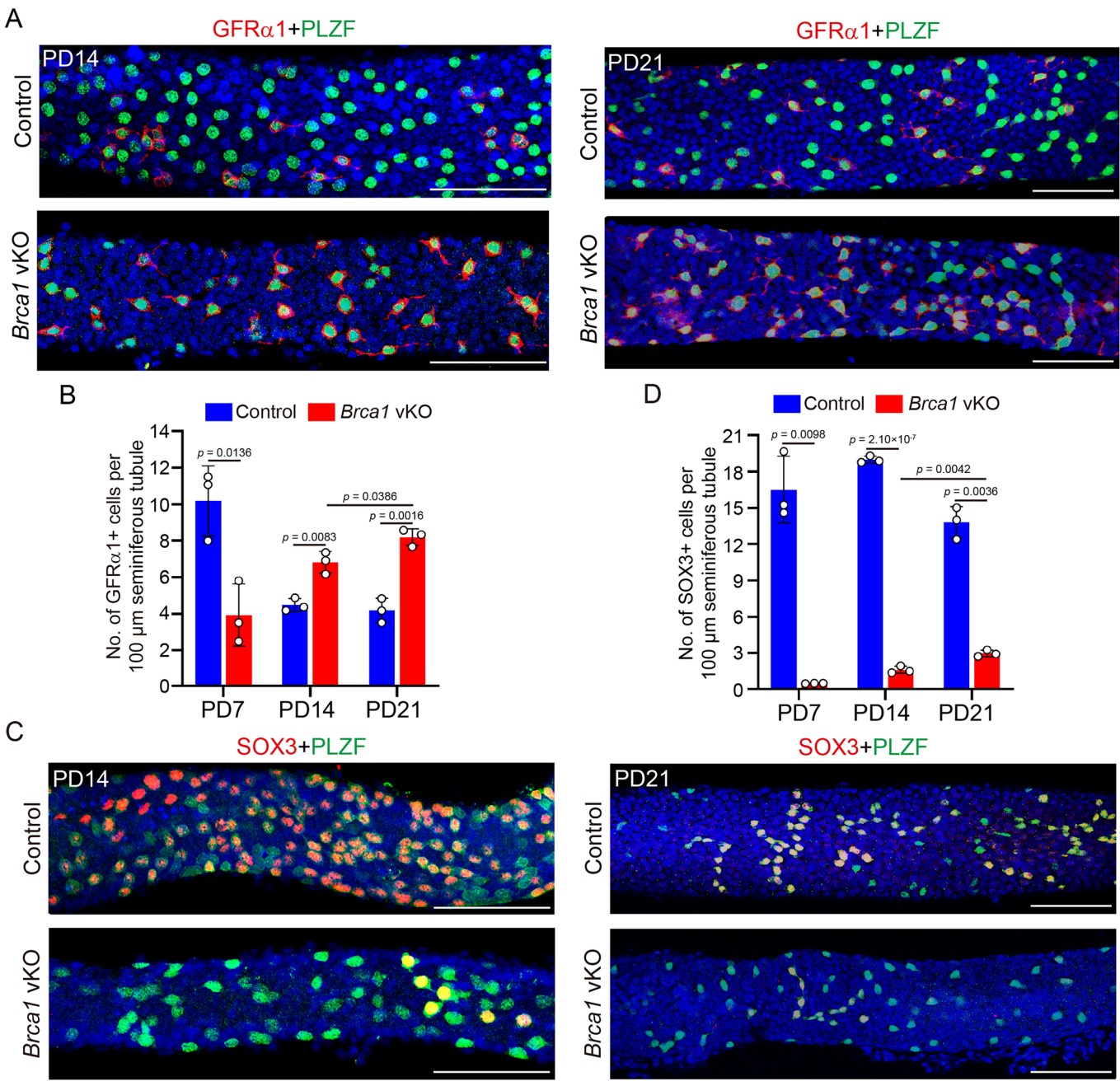

**Figure EV5. The number of spermatogonial stem cells gradually increases in *Brca1* vKO testes.**

(**A**) Whole-mount IF staining of GFRα1 and PLZF in seminiferous tubules of testes from Control and *Brca1* vKO male mice at PD14 and PD21. (**B**) Statistical analysis of the number of GFRα1+ cells per 100 μm seminiferous tubule in testes from Control and *Brca1* vKO male mice at PD7, PD14, and PD21. Three mice of each genotype were analyzed. (**C**) Whole-mount IF staining of SOX3 and PLZF in seminiferous tubules of testes from Control and *Brca1* vKO male mice at PD14 and PD21. (**D**) Statistical analysis of the number of SOX3+ cells per 100 μm seminiferous tubule in testes from Control and *Brca1* vKO male mice at PD7, PD14, and PD21. Three mice of each genotype were analyzed. Data information: Data are presented as mean ± SD. *P* value, two-tailed unpaired Student's *t* test. Scale bar, 100 μm.

