## [Peer Review File · EMBO Reports]

BRCA1 preserves genome integrity during the formation of undifferentiated spermatogonia

Peng Li, Licun Song, Longfei Ma, Chunsheng Han, Lejun Li, Lin-Yu Lu, and Yidan Liu

Corresponding author(s): Yidan Liu (yidanliu@zju.edu.cn), Lejun Li (lilejun@zju.edu.cn), Lin-Yu Lu (lulinyu@zju.edu.cn)

Review Timeline:

Submission Date:	29th Oct 24
Editorial Decision:	9th Dec 24
Revision Received:	29th Mar 25
Editorial Decision:	30th Apr 25
Revision Received:	9th May 25
Accepted:	16th May 25

Editor: Achim Breiling

Transaction Report:

Dear Dr. Liu,

Thank you for the transfer of your manuscript to EMBO reports. I have now received the reports from the three referees that were asked to evaluate your study, which can be found at the end of this email.

As you will see, the referees think that these findings are of interest. However, they have several comments, concerns, and suggestions, indicating that a major revision of the manuscript is necessary to allow publication of the study in EMBO reports. As the reports are below, and all the referee concerns need to be addressed, I will not detail them here.

Given the constructive referee comments, I would like to invite you to revise your manuscript with the understanding that the concerns of the referees must be addressed in the revised manuscript and in a detailed point-by-point response. Acceptance of your manuscript will depend on a positive outcome of a second round of review. It is EMBO reports policy to allow a single round of revision only and acceptance of the manuscript will therefore depend on the completeness of your responses included in the next, final version of the manuscript.

- 1) a .docx formatted version of the final manuscript text (including legends for main figures, EV figures and tables), but without the figures included. Figure legends should be compiled at the end of the manuscript text.
- 2) individual production quality figure files as .eps, .tif, .jpg (one file per figure), of main figures and EV figures. Please upload these as separate, individual files upon re-submission.

- 4) a complete author checklist, which you can download from our author guidelines (<https://www.embopress.org/page/journal/14693178/authorguide>). Please insert page numbers in the checklist to indicate where the requested information can be found in the manuscript. The completed author checklist will also be part of the RPF.

- 5) that primary datasets produced in this study (e.g. RNA-seq, ChIP-seq, structural and array data) are deposited in an

appropriate public database. If no primary datasets have been deposited, please also state this in a dedicated section (e.g. 'No primary datasets have been generated and deposited'), see below.

The accession numbers and database should be listed in a formal "Data Availability" section that follows the model below. This is now mandatory (like the COI statement). Please note that the Data Availability Section is restricted to new primary data that are part of this study. This section is mandatory. As indicated above, if no primary datasets have been deposited, please state this in this section

Data availability

8) Regarding data quantification and statistics, please make sure that the number "n" for how many independent experiments were performed, their nature (biological versus technical replicates), the bars and error bars (e.g. SEM, SD) and the test used to calculate p-values is indicated in the respective figure legends (also for EV and Appendix figures). Please also check that all the p-values are explained in the legend, and that these fit to those shown in the figure. Please provide statistical testing where applicable. Please avoid the phrase 'independent experiment', but clearly state if these were biological or technical replicates. Please also indicate (e.g. with n.s.) if testing was performed, but the differences are not significant. In case n=2, please show the data as separate datapoints without error bars and statistics. See also: <http://www.embopress.org/page/journal/14693178/authorguide#statisticalanalysis>

9) Please add scale bars of similar style and thickness to microscopic images, using clearly visible black or white bars (depending on the background). Please place these in the lower right corner of the images themselves. Please do not write on or near the bars in the image but define the size in the respective figure legend.

10) Please also note our reference format:

12) We now use CRedit to specify the contributions of each author in the journal submission system. CRedit replaces the author contribution section. Please use the free text box to provide more detailed descriptions and do NOT provide your final manuscript text file with an author contributions section. See also our guide to authors: <https://www.embopress.org/page/journal/14693178/authorguide#authorshipguidelines>

13) All Materials and Methods need to be described in the main text using our 'Structured Methods' format, which is required for

all research articles. According to this format, the Methods section should include a Reagents and Tools Table (listing key reagents, experimental models, software, and relevant equipment and including their sources and relevant identifiers), uploaded as separate file, and a Methods section in which we encourage the authors to describe their methods using a step-by-step protocol format with bullet points, to facilitate the adoption of the methodologies across labs. More information on how to adhere to this format as well as downloadable templates (.doc) for the Reagents and Tools Table can be found in our author guidelines (section 'Structured Methods'):

14) Please add up to 5 keywords to the manuscript and order the sections like this, using these names:

Title page - Abstract - Keywords - Introduction - Results - Discussion - Methods - Data availability section - Acknowledgements (including funding information) - Disclosure and Competing Interests Statement - References - Figure legends - Expanded View Figure legends

15) Please make sure that all the funding information is also entered into the online submission system and that it is complete and similar to the one in the acknowledgement section of the manuscript text file.

I look forward to seeing a revised form of your manuscript when it is ready.

Yours sincerely,

Referee #1:

In this manuscript, the authors investigated the role of BRCA1 in regulating homeostasis of murine spermatogonial stem cells, using a germline specific BRCA1-inducible knock-out mutant. To start with, the authors showed that DNA damage is widespread during early post-natal maturation phase of gonocytes to spermatogonial stem cells (SPG) (@PD1-PD14) in WT mice. Using a using a germ cell-specific Brca1 KO and scRNA transcriptomics, they show a global loss of SPG in the mutant and pinpoint a population of progenitor cells (SPG2) that are particularly sensitive to the loss of Brca1. Looking a relative proportion of SPG populations, the authors then suggest that mutant spermatogonial stem cells (unexpectedly) hyper-proliferate, resulting in their premature exhaustion and loss of fertility. Finally, the authors show evidence suggesting that formaldehyde is the likely damaging agent operating during spermatogonial stem cell maturation, explaining why this developmental stage requires an active DNA damage repair machinery to support fertility later during adulthood.

Overall, this manuscript follows a logical flow of enquiry and is supported by relevant experiments to assess the role of Brca1, and more generally DNA damage repair, in the generation of postnatal undifferentiated spermatogonial stem cells. However, I have some concerns about the lack of explanation for choosing specific time points for analysis and more generally the lack of clarity of the narrative that does not specify and distinguish statements relating to mice vs. humans (nor for 'infantile vs. adult spermatogenesis), or the generalisability of the findings.

Specific comments include

1. Based on the timepoints (PD1-PD14) chosen for the experimental design, I have had to infer that the authors are investigating a specific stage of male germ cells development, the so-called "first wave of spermatogenesis", a unique phase of undifferentiated spermatogonial stem cell maturation that does not lead to production of sperm. Whilst this process is important to set up "adult spermatogenesis" in mice, it is distinct from 'homeostasis' and maintenance of spermatogonial stem cells in adults. This will need to be clarified throughout the text.

Moreover, this "first wave of spermatogenesis" does not occur in humans and caution must be taken when extending the results to humans. For example, in lines 369 to 370, the sentence is in reference to spermatogenesis in humans. However, the authors discuss their results to suggest that the phenomenon observed in this manuscript would also impact human fertility. Of note, the introductory paragraph needs some attention as currently a mixed description of mouse and human spermatogenesis is provided. At the very least, the authors need to specify when they refer to mouse vs. human spermatogenesis. They should also mention that the mouse model is a homozygous KO, a situation that can happen in humans, but would be rather unusual.

2. Precise description of the experimental timing is important for the interpretation of the data and the timepoints chosen for analysis will need to be justified better. For example, on line 135, the authors state "A previous study of gene expression profiles of undifferentiated spermatogonia in infant and adult mice has revealed that the expressions of genes in BRCA1-related DNA damage response pathways are higher in PD6 than in adults"- This begs the question as why do they chose to do their transcriptomics analysis only at PD7 and not earlier/later on?

Line 141: Please state the vasa-Cre is active from ~E15 and how this is relevant to the short period of time they studied (PD1-PD14).

Line 147: Although the testes of Brca1 vKO male mice were indistinguishable from those of control mice at PD1, they were significantly smaller than control mice at PD7 (Figure 2B-C). What happens to these testes in the adult or at PD14?

3. Line 149: scTranscriptomics is based on a single pair of control and Brca1 vKO at PD7, presumably from a single mouse - if it is the case, this is not ideal and several mice (or biological replicates) should have been included?

It is difficult to understand why in this context, the Brca1 vKO testes (which are smaller and have fewer germ cells), ended up with almost twice as many 'high quality' cells (5077 cells from control testes and 8711 cells from mutant testes)?

Could you explain these numbers. I am also concerned that the analysis is entirely based on relative proportion of germ cells (SPG1-2-3, in respect to the other somatic cells) and for the mutant the number of mutant SPG is extremely small.

Sup Figure 4 - the identity of SPG2 is not well defined based on gene expression markers. For example Sox3 and nanos3 are expressed across 2 clusters.

4. A large proportion of the results in this manuscript consists of antibody staining of seminiferous tubules and quantification of positively stained cells. While this is good practice it will be essential to explain better how this was performed. I would advice that the authors provide a table with their quantification data and the level of statistical significance of differences - relative proportion can be very problematic and difficult to interpret, especially for small numbers.

Specific (not exhaustive) examples of issues are listed below

- For example, the choice of markers used is not always consistent. I note that figure 1 uses PLZF as a marker of undifferentiated SPG, while GFRa1 (which is used later on) would have been a more natural choice. I also note that PLZF is not described in Sup Fig4. In Fig 5D, can the authors explain to the reader the benefit of the using MVH as a marker, as opposed to using PLZF or GFRa1 for germ cells?

- Moreover, it is unclear how positive cells were actually quantified (i.e. I have tried to find the information as to whether counting is in cross-sections or across the tubule length, how many tubules/how many mice, but this is not available in the methods or in the fig legend). As spermatogenesis occurs in a wave-like progression within seminiferous tubules (PMID: 16540512), how the cells are counted will depends on the stage of spermatogenic wave the tubules is examined.

- I note for example that in Fig 5A, the number of GFRa1 cells appears to be similar between wild-type control and Brca1 vKO mice, however, the quantification of the numbers are different in Fig 5B. How were GFRa1 cells quantified, i.e. across the length of a tubule, or on a cross-section?

- Additionally, it is surprisingly that at PD180 (Fig 5B), there is little GFRa1 cells per tubules in wild-type, which suggests that the WT mice are no longer fertile

- Can the authors explain why in Fig 4G that the measurement of GFRa1 and Sox3 is normalised to PLZF cells and not in previous analyses?

- In Fig 7C and 7E, it appears that proportionally, there are few cells that are double-stained for both GFRa1 (or sox3) with γ H2AX in Brca1 vKO tubules, which is not the case in Fig 7D and 7F, respectively. As with before, can the authors explain how the cells were quantified?

- Can the authors explain why BrdU was chosen in Fig 8A as a marker for replicating cells, whilst in previous figures, p-H3(S10) was used instead?

- For Fig 8C, the authors used PLZF as a measurement of undifferentiated spermatogonial stem cells, which is inconsistent with the markers with previous quantifications. Similarly, in Fig 8G, the authors used MVH as a measure of germ cells, which is not consistent with previous experiments.

5. One of the main finding of the paper is the unexpected hyper-proliferation of SPG1 in the Brca1-vko at PD7 (Figure 4A). This interpretation is based on the relative proportion of the staining of p-H3(S10) vs. GFRa1 (Figure 4A).

First, based on Fig4A alone it is difficult to be convinced by this because the quality p-H3(S10) stain is not very convincing (i.e. I note the staining is clearer in Fig 7G) and (as point above) it is unclear how many tubules/mice were analysed. This is an important conclusion, and the quantitative data will need to be provided as a table to support Figure 4B. Additionally, data collected across multiple timepoints (PD14 and PD21), shown in the suppl figs 4-5, appear to show reduced undifferentiated spermatogonial stem cells in the mutant mouse model, compared to the controls. My interpretation of the data suggests that "hyper-proliferation" of SPG1 stalls over time. Could you comment? I was also surprised to see that the number of GFRa1 cells

drops by half between PD7 vs PD14/PD21 in wild-type control. Can the authors explain how they have normalised the count of GFRa1 cells in sup Fig 5B, as there is more cells per 100 μm of tubules than observed in Fig 4D. Furthermore, there is contradiction in the data shown between Fig 4D and sup Fig 5B, which shows (for the wild-type control) that there is only a decrease in GFRa1 cells between PD14 and PD21 (sup Fig 5B), whilst the number is halved (as previously stated). This is in contrast to the measurement of SOX3 cells at the same time points between Fig 4F and sup Fig 5D.

6. In line 259, the authors stated: preventing progenitor spermatogonia's death should restore the homeostasis of undifferentiated spermatogonia and prevent the hyper-proliferation of spermatogonial stem cells. It is not evident earlier in the manuscript if cell death is different wild-type, compared to Brca1 vKO mice. Could you comment
Note that it has also been documented that during the first wave of spermatogenesis, spermatogonial stem cells undergo a high frequency of apoptosis (PD8-PD22), compared to adult spermatogonial stem cells (PMID: 8989527, 20403866, 8674410). As such, this might be another possible source of DNA damage. Additionally, in another Brca1 knock-out mouse model, increased apoptosis in the mutant has been described (PMID: 12642502).

Minor comments:

I have the following queries regarding the results:

1. Please specify in first section of the results (lines 97 to 130) that the experiments were performed on wild-type control mice.
2. In reference to figure 2G, there should be a p-value if the authors claim that there is a significant decrease in the percentage of germ cells.
3. In line 181, the authors stated that the percentage ratio of SPG3 and SPG2 was similar between wild-type control and Brca1 vKO. Is 0.61 (wild-type) 'similar' to 0.98 (Brca1 vKO), as shown in figure 3C?
4. For supplementary figure 4A, it is unclear in the figure legend if the t-SNE plot is in reference to all germ cells (control + Brca1 vKO), or only from control germ cells.
5. Figure 4D & 4F: Is the increase observed in Brca1 vKO mice for GFRa1+, or SOX3+, cells significant between PD14 and PD21?
6. In figure 6D, when the number of GFRa1+ and SOX3+ cells were normalised to percentage of PLZF cells, they appear similar. However, raw SOX3+ cell count per tubules in figure 6C suggests that there is still a significant reduction between wild-type control and Brca1-p53 vDKO mice. Do these results mean that there is an overall reduced number of PLZF cells in the Brca1-p53 vDKO mouse model? Can the authors show a quantification of PLZF cells in the Brca1-p53 vDKO mouse model compared to wild-type control?
7. In the same reference to figure 6D and in regards to lines 268 - 269, if there is no difference observed for either GFRa1+, or SOX3+, cells between wild-type control and Brca1-p53 vDKO mice, statistical analyses should be presented in the figure (i.e. ns).
8. For the same staining of p-H3(S10) in the tubules, there is a difference in the number of positively stained cells between Figure 7G (top), compared to Figure 4A (top).
9. The authors hypothesised that increased proliferation with defective DNA repair leads to elevated cell death (lines 309 to 312). The manuscript will be improved if there is supporting evidence of cleaved PARP1 measurement (or a quantitative measurement of cell death) in the seminiferous tubules of their Brca1 vKO mouse model.
10. It is not clearly written in lines 321 to 323 that the experiments were performed on wild-type mice. This information was only inferred after looking at figure 8A.
11. In line 329, the authors stated that the BrdU-stained cells are likely Sertoli cells. What is the evidence to support this statement.
12. The authors only presented an IF staining for figure 8G, but have not quantified the number of MVH+ cells (as a separate figure).
13. Figure 9 (the model) is not referenced in text. Please also add "Timepoints" to the description of this model.

Typographical errors and corrections

14. In general, authors should list the number of replicates in figure legend. Typo in figure legend of supplementary figure 4E. It should be SOX3+, instead of GFRa1+.
15. In figure 5E, a scale bar is missing for control PD90 image.
16. Check that murine 'BRCA1' is written as Brca1 throughout the text

Referee #2:

This manuscript, authored by Li et al., investigates the impact of germline-specific conditional knockout of BRCA1 on the cellular dynamics of spermatogonia in mice. BRCA1, an ubiquitin ligase, is involved in various DNA repair pathways. Previous studies have shown that complete BRCA1 knockout leads to embryonic lethality, and truncated hypomorphic alleles, conditional knockouts, and compound mutations (e.g., with p53) have revealed the role of BRCA1 in spermatogenesis, ovarian reserve, and meiosis.

In this study, the authors re-examine spermatogenesis in male mice with a germline-specific Brca1 knockout, focusing on the number of spermatogonial stem cells (SSCs) and progenitor spermatogonia using detailed histological analysis. A key finding is

that Brca1 loss-of-function in the male germline results in a more severe reduction in the number of progenitor spermatogonia compared to SSCs. Another observation is that SSCs in neonatal testes transiently exhibit strong gH2AX staining, which is absent in adult testes. The authors suggest that gH2AX staining indicates higher levels of DNA damage in neonatal SSCs. However, alternative interpretations of gH2AX staining should be considered. Additionally, the authors propose that formaldehyde, endogenously generated in neonatal SSCs, may contribute to DNA damage, based on the observation that the number of spermatogonia in the Brca1 and Aldh2 double knockout is reduced more severely than in the single knockouts. This interpretation again requires more careful consideration of the data.

While the manuscript provides potentially interesting insights into the regulation of genome integrity in SSCs, the causal relationships among the three main findings (1) the reduced number of neonatal SSCs in Brca1 knockout, (2) the increased gH2AX staining in neonatal SSCs, and (3) the proposed role of formaldehyde, are not experimentally clear. Additional data are needed to support the authors' hypothesis. My major points are as follows:

Major Points:

1. Interpretation of gH2AX signals

gH2AX signals increase not only in response to DNA double-strand breaks (DSBs) but also due to chromatin conformation changes. Actually, drastic epigenetic changes should occur in neonatal spermatogonial stem cells. To distinguish between DSBs and chromatin changes, additional data on DSBs are needed. This could include immunostaining for other DNA damage markers (e.g., 53BP1), the comet assay, or DNA damage sequencing.

2. Quantification of gH2AX foci

gH2AX typically forms nuclear foci, and the number of these foci is commonly quantified to assess DNA damage. It is unclear why the standard quantification of gH2AX foci was not performed in this study. If BRCA1 functions to repair DNA damage detected by gH2AX in spermatogonia, one would expect the signals or foci of gH2AX (and other DNA damage markers) to be higher in Brca1 KO testes compared to controls. However, the authors did not present such data. More careful experiments are needed to support the claim that "BRCA1 preserves genome integrity during the formation of undifferentiated spermatogonia," as stated in the manuscript's title.

3. Differentiation of spermatogonia

It is known that the differentiation process of spermatogonia during the first wave of spermatogenesis in neonatal testes is distinct from that in adult testes. At postnatal days 4-5 (PD4-5), spermatogonial stem cells likely begin to differentiate. Therefore, a potential correlation between spermatogonial differentiation (toward differentiating spermatogonia and meiosis) and increased gH2AX signals in neonatal testes should be carefully examined and discussed.

4. Clarification of DNA Damage generation and repair

In line 304, the authors state, "since DNA damage generation and repair deficiency were similar between these two populations in Brca1 vKO testes." However, how can DNA damage generation and repair be inferred solely from gH2AX staining? Further clarification or additional experiments are needed to support this claim.

5. Proliferation vs. cell death

In line 309, the authors state, "since cell proliferation in the presence of unrepaired DNA damage is detrimental, the higher proliferation rate may lead to elevated cell death in progenitor spermatogonia when spontaneous DNA damage fails to be repaired efficiently in the absence of BRCA1." I agree with the general concept that highly proliferating cells are more susceptible to DNA damage. However, if this is true, why do the hyperproliferating spermatogonial stem cells in Brca1 KO not undergo more cell death, but instead continue to hyperproliferate? This discrepancy should be addressed.

6. Lack of cell death analysis

While the manuscript includes several studies on cell proliferation kinetics, only a single panel (Figure 1D) is presented for cell death. A more thorough investigation of cell death, which is crucial for interpreting the data, would strengthen the manuscript.

7. Single cell RNA sequencing (scRNAseq) data

The authors use scRNAseq to quantify the number and proportion of spermatogonia (SPG1-3). However, no mention is made of potential differences in gene expression, such as those downstream of the DNA damage checkpoint or other relevant pathways. This should be addressed.

8. Brca1 and Aldh2 double knockout

The authors present data showing that the number of spermatogonia is more severely reduced in Brca1 and Aldh2 double knockout testes compared to each single knockout. However, gH2AX and other DNA damage markers (including signal intensities and nuclear foci) are not examined. It is unclear whether "formaldehyde is likely a source of spontaneous DNA damage during the formation of undifferentiated spermatogonia" can be concluded from simply counting the number of cells. Further analysis (to establish cause and effect) of DNA damage in the double knockout model would be necessary to support this hypothesis.

Referee #3:

This is an interesting paper creating insights into the establishment of spermatogonial stem cell populations. The paper uses elegant and up-to-date technologies and presents valid datasets.

The authors report somewhat surprising findings on DNA damage of spermatogonial subpopulations by thus far neglected mechanisms.

The paper provides significant and thoughtful new insights. I consider the discussion too long and slightly unfocused.

Otherwise the studies are well performed, the paper is carefully designed and the data are well presented.

Referee #1:

In this manuscript, the authors investigated the role of BRCA1 in regulating homeostasis of murine spermatogonial stem cells, using a germline specific BRCA1-inducible knock-out mutant. To start with, the authors showed that DNA damage is widespread during early post-natal maturation phase of gonocytes to spermatogonial stem cells (SPG) (@PD1-PD14) in WT mice. Using a using a germ cell-specific Brca1 KO and scRNA transcriptomics, they show a global loss of SPG in the mutant and pinpoint a population of progenitor cells (SPG2) that are particularly sensitive to the loss of Brca1. Looking a relative proportion of SPG populations, the authors then suggest that mutant spermatogonial stem cells (unexpectedly) hyper-proliferate, resulting in their premature exhaustion and loss of fertility. Finally, the authors show evidence suggesting that formaldehyde is the likely damaging agent operating during spermatogonial stem cell maturation, explaining why this developmental stage requires an active DNA damage repair machinery to support fertility later during adulthood.

Overall, this manuscript follows a logical flow of enquiry and is supported by relevant experiments to assess the role of Brca1, and more generally DNA damage repair, in the generation of postnatal undifferentiated spermatogonial stem cells. However, I have some concerns about the lack of explanation for choosing specific time points for analysis and more generally the lack of clarity of the narrative that does not specify and distinguish statements relating to mice vs. humans (nor for 'infantile vs. adult spermatogenesis), or the generalisability of the findings.

Thank you for your positive feedback on our study. We have addressed the concerns raised by conducting additional experiments and analyses. We truly appreciate your detailed suggestions, which have been invaluable in improving the quality of our manuscript.

Specific comments include

1. Based on the timepoints (PD1-PD14) chosen for the experimental design, I have had to infer that the authors are investigating a specific stage of male germ cells development, the so-called "first wave of spermatogenesis", a unique phase of undifferentiated spermatogonial stem cell maturation that does not lead to production of sperm. Whilst this process is important to set up "adult spermatogenesis" in mice, it is distinct from 'homeostasis' and maintenance of spermatogonial stem cells in adults. This will need to be clarified throughout the text.

We have clarified in the text that our study focuses on the formation of undifferentiated spermatogonia that occur during PD1-PD14 in mice, that is different from homeostasis and maintenance of spermatogonial stem cells in adults (line 91-98).

Moreover, this "first wave of spermatogenesis" does not occur in humans and caution must be taken when extending the results to humans. For example, in lines 369 to 370, the sentence is in reference to spermatogenesis in humans. However, the authors discuss their results to suggest that the phenomenon observed in this manuscript would also impact human fertility.

Of note, the introductory paragraph needs some attention as currently a mixed description of mouse and human spermatogenesis is provided. At the very least, the authors need to specify when they refer to mouse vs. human spermatogenesis. They should also mention that the mouse model is a homozygous KO, a situation that can happen in humans, but would be rather unusual.

We agree that our findings in mice may not be directly applicable to humans. To address this, we have revised the introduction and discussion to focus exclusively on mouse spermatogenesis (line 47-67, 437-448).

2. Precise description of the experimental timing is important for the interpretation of the data and the timepoints chosen for analysis will need to be justified better.

We agree. We have revised the manuscript to more precisely describe the experimental timing and provide a clearer justification for the time points chosen for analysis.

For example, on line 135, the authors state "A previous study of gene expression profiles of undifferentiated spermatogonia in infant and adult mice has revealed that the expressions of genes in BRCA1-related DNA damage response pathways are higher in PD6 than in adults"- This begs the question as why do they chose to do their transcriptomics analysis only at PD7 and not earlier/later on?

We begin our study by collecting testes at various timepoints to analyze the timing of undifferentiated spermatogonia formation and the dynamics of DNA damage during this process. DNA damage in undifferentiated spermatogonia is high at PD4 and PD7, but diminishes at PD14, suggesting that DNA damage repair starts before PD7. To gain further insight into DNA damage repair in undifferentiated spermatogonia before PD7, we revisited previous scRNA-Seq analyses from earlier timepoints. Specifically, scRNA-Seq data at PD6 suggest that BRCA1 may play a key role in this process.

Upon generating *Brcal* vKO mice, we observed that their testes are smaller than those of control mice at PD7, suggesting that BRCA1 loss impairs germ cell development before PD7. Given that undifferentiated spermatogonia formation is not yet complete at PD4 but nearly finished by PD7 in WT mice, we chose PD7 as the optimal timepoint to analyze the impact of BRCA1 loss on undifferentiated spermatogonia formation. Therefore, we performed scRNA-Seq analysis at PD7.

We have included these descriptions in the revised manuscript (line 167-170).

Line 141: Please state the vasa-Cre is active from ~E15 and how this is relevant to the short period of time they studied (PD1-PD14).

In *Vasa-Cre* mice, the Cre recombinase starts to express in gonocytes at E15.5, which depletes BRCA1 before the formation of undifferentiated spermatogonia from PD1 to PD14, making *Brca1^{fl/fl} Vasa-Cre* (*Brca1* vKO) a suitable mouse model to study the impact of BRCA1 loss on undifferentiated spermatogonia formation. We have confirmed that BRCA1 was undetectable in germ cells when undifferentiated spermatogonia start to form at PD1 (Figure 2A). We have included these descriptions in the revised manuscript (line 159-163).

Line 147: Although the testes of Brca1 vKO male mice were indistinguishable from those of control mice at PD1, they were significantly smaller than control mice at PD7 (Figure 2B-C). What happens to these testes in the adult or at PD14?

We have thoroughly compared the sizes and weights of control and *Brca1* vKO testes at different timepoints. The *Brca1* vKO testes were smaller than control testes at all timepoints beyond PD7, including PD14 and adults (Figure EV3D-E). We have included these descriptions in the revised manuscript (line 164-166).

3. Line 149: scTranscriptomics is based on a single pair of control and Brca1 vKO at PD7, presumably from a single mouse - if it is the case, this is not ideal and several mice (or biological replicates) should have been included?

We acknowledge that including multiple biological replicates would have been ideal for the scRNA-Seq analyses. At the time of planning this study several years ago, the relatively high cost of scRNA-Seq analyses limited our ability to include additional samples. As a result, this experiment was designed as a pilot to provide preliminary insights. The observations from the scRNA-Seq analyses were subsequently validated using immunofluorescence staining with multiple markers and an appropriate number of replicates. We hope this approach mitigates the limitations of the initial experimental design.

It is difficult to understand why in this context, the Brca1 vKO testes (which are smaller and have fewer germ cells), ended up with almost twice as many 'high quality' cells (5077 cells from control testes and 8711 cells from mutant testes)?

Could you explain these numbers. I am also concerned that the analysis is entirely based on relative proportion of germ cells (SPG1-2-3, in respect to the other somatic cells) and for the mutant the number of mutant SPG is extremely small.

During scRNA-Seq, the total number of cells captured is determined by the number of cells loaded onto the sequencer. During our experiment, we observed that the testes

from *Brcal* vKO mice were significantly smaller than those from control mice at PD7, suggesting that BRCA1 loss impairs germ cell development (Figure 2C). To mitigate the potential issue of capturing an insufficient number of germ cells in *Brcal* vKO samples, we increased the number of cells from *Brcal* vKO that was loaded onto the sequencer. As a result, the number of cells captured from the control testes was smaller than that from *Brcal* vKO testes.

However, it is important to note that the number of cells captured does not directly correspond to the absolute cell number in a sample, as scRNA-Seq analysis employs a sampling approach, randomly selecting a subset of cells to represent the entire population. Due to this sampling nature, our analysis is based on the relative proportion rather than the exact numbers. Despite capturing more cells, the proportion of germ cells in *Brcal* vKO testes was dramatically reduced. This finding aligns with our conclusion that the germ cell population is decreased in *Brcal* vKO testes.

We hope this explanation clarifies the observed results.

Sup Figure 4 - the identity of SPG2 is not well defined based on gene expression markers. For example Sox3 and nanos3 are expressed across 2 clusters.

The cell clusters were generated in a unbiased manner during the scRNA-Seq analyses. To cluster the germ cell population, we first identified the top 1000 unbiased variable features with high variability across cells using the Seurat-embedded function 'FindVariableFeatures'. Dimensional reduction was then applied based on the variable features, followed by clustering analysis using a graph-based approach. Importantly, no pre-selected gene markers were used during the above steps.

To define the identities of the unbiased germ cell clusters, we subsequently performed gene expression plots of canonical germ cell markers, referencing multiple previous studies on mouse spermatogenesis. As is typical with unbiased clustering, some genes are expressed in multiple clusters. For example, in our study, Nanos3 and Sox3 are expressed in two clusters, a pattern also observed in previous scRNA-Seq analyses (Dura *et al.*, 2022; Wang *et al.*, 2019). However, the identity of each unbiased cluster is defined by the expression of a combination of genes, not by individual genes that might express in multiple clusters. Supporting this, the identities of the germ cell clusters identified in our study align with those from previous studies (Dura *et al.*, 2022; Wang *et al.*, 2019).

We hope this clarifies the observed expression patterns and the methodology used in our scRNA-Seq analyses.

4. A large proportion of the results in this manuscript consists of antibody staining of seminiferous tubules and quantification of positively stained cells. While this is good practice it will be essential to explain better how this was performed. I would advice

that the authors provide a table with their quantification data and the level of statistical significance of differences - relative proportion can be very problematic and difficult to interpret, especially for small numbers.

We have included the quantification data used for calculating the relative proportions as well as the level of statistical significance of differences for all the relevant figures in Table EV1.

Specific (not exhaustive) examples of issues are listed below

- For example, the choice of markers used is not always consistent. I note that Figure 1 uses PLZF as a marker of undifferentiated SPG, while GFR α 1 (which is used later on) would have been a more natural choice.

PLZF is a marker for all undifferentiated spermatogonia, which consist of two subpopulations: GFR α 1-positive spermatogonial stem cells and SOX3-positive progenitor spermatogonia. In Figure 1, as part of the initial characterization of γ H2AX signals in undifferentiated spermatogonia at different timepoints, we used PLZF to label all undifferentiated spermatogonia without distinguishing between the two subpopulations. After scRNA-seq analyses suggested that progenitor spermatogonia might be more susceptible than spermatogonial stem cells to BRCA1 loss, we used GFR α 1 and SOX3 to label the two subpopulations and examined the γ H2AX signals in each of them.

I also note that PLZF is not described in Sup Fig4.

We have now included the quantification data for PLZF+ undifferentiated spermatogonia in Figure EV4 (original Supplementary Figure 4) and described it in the revised manuscript (line 211-212).

In Fig 5D, can the authors explain to the reader the benefit of using MVH as a marker, as opposed to using PLZF or GFR α 1 for germ cells?

In addition to PLZF and GFR α 1 (Figure 5A-C), we initially used the germ cell marker MVH as another parameter to evaluate undifferentiated spermatogonia in *Brca1* vKO testes at different time points (original Figure 5D). However, we realize that, unlike in *Brca1* vKO testes, undifferentiated spermatogonia are not the only germ cell population in control testes, which also contain other germ cell types, including meiotic cells. Given this, we recognize that MVH is not suitable for comparing the number of undifferentiated spermatogonia between control and *Brca1* vKO mice. To avoid any potential misinterpretation, we have removed the MVH data (original Figure 5D) from the manuscript.

- Moreover, it is unclear how positive cells were actually quantified (i.e. I have tried to find the information as to whether counting is in cross-sections or across the tubule

length, how many tubules/how many mice, but this is not available in the methods or in the fig legend). As spermatogenesis occurs in a wave-like progression within seminiferous tubules (PMID:16540512), how the cells are counted will depends on the stage of spermatogenic wave the tubules is examined.

The quantification was performed on both cross-sections and isolated seminiferous tubules. For cross-sections, the positive cells in at least 20 intact seminiferous tubules were counted. The number of positive cells per tubule was calculated by dividing the total number of positive cells by the number of seminiferous tubules analyzed. The data obtained from 3 mice were presented.

For isolated seminiferous tubules, the positive cells were counted in seminiferous tubules that are at least 1 mm by length. The number of positive cells per 100 μm seminiferous tubule was calculated by dividing the total number of all positive cells by the length of seminiferous tubules. The numbers obtained from 3 mice were presented.

We have now included this information in Methods part of the revised manuscript (line 597-600, 619-624).

- I note for example that in Fig 5A, the number of GFR α 1 cells appears to be similar between wild-type control and Brca1 vKO mice, however, the quantification of the numbers are different in Fig 5B. How were GFR α 1 cells quantified, i.e. across the length of a tubule, or on a cross-section?

The quantification of GFR α 1-positive cells was performed on cross-sections using the same method described above (line 597-600). We noticed that the clarity of GFR α 1 signals was suboptimal in the merged GFR α 1 and PLZF images. To address this, we have included images showing GFR α 1 signals alone, which clearly highlight the difference in the number of GFR α 1-positive cells between control and Brca1 vKO mice, consistent with the quantification data in Figure 5B.

- Additionally, it is surprisingly that at PD180 (Fig 5B), there is little GFR α 1 cells per tubules in wild-type, which suggests that the WT mice are no longer fertile

After the establishment of undifferentiated spermatogonia at PD14, GFR α 1-positive spermatogonial stem cells only constitutes around 20% of all PLZF-positive undifferentiated spermatogonia (Figure EV1E). Consistently, the numbers of GFR α 1-positive spermatogonial stem cells are much fewer than PLZF-positive undifferentiated spermatogonia on cross-sections of mouse testes at all timepoints analyzed after PD14, including PD180. The rarity of GFR α 1-positive spermatogonial stem cells is consistent with the stem cell nature and the relatively low proliferation rate of these cells.

Consistent with the general idea that the reproductive span of male mice lasts for more than 1 year, the numbers of GFR α 1-positive spermatogonial stem cells and the all PLZF-positive undifferentiated spermatogonia per tubule do not drop dramatically from PD21 to PD180. Therefore, the WT male mice analyzed in our study should still be fertile at PD180.

- Can the authors explain why in Fig 4G that the measurement of GFR α 1 and Sox3 is normalised to PLZF cells and not in previous analyses?

PLZF is a marker for all undifferentiated spermatogonia, which consist of two subpopulations: GFR α 1-positive spermatogonial stem cells and SOX3-positive progenitor spermatogonia. Analyzing the percentages of two populations among all PLZF-positive undifferentiated spermatogonia can reveal the composition of undifferentiated spermatogonia.

In Figure EV1 (original Supplementary Figure 1), we have performed this analysis at different timepoints during the establishment of undifferentiated spermatogonia in WT mice. Starting from PD1, the percentage of GFR α 1-positive spermatogonial stem cells gradually decreases and that of SOX3-positive progenitor spermatogonia gradually increases. The percentages of these two populations become steady at PD14, suggesting that these two populations have reached homeostasis.

In Figure 3F, we have performed this analysis and found that the percentage of SOX3-positive progenitor spermatogonia dramatically decreases and that of GFR α 1-positive spermatogonial stem cells dramatically increases in *Brcal* vKO mice at PD7, revealing a significant delay toward the homeostasis between these two populations.

In Figure 4G, we also performed this analysis and found that the percentages of these two populations at PD14 and PD21 are similar to those at PD7 in *Brcal* vKO mice. Therefore, unlike in control mice, the homeostasis of between these two populations are never established in *Brcal* vKO mice, despite a significant increase in the numbers of GFR α 1-positive cells.

*- In Fig 7C and 7E, it appears that proportionally, there are few cells that are double-stained for both GFR α 1 (or sox3) with γ H2AX in *Brcal* vKO tubules, which is not the case in Fig 7D and 7F, respectively. As with before, can the authors explain how the cells were quantified?*

The numbers of the cells positive for GFR α 1/SOX3 and those double-positive for GFR α 1/SOX3 and γ H2AX are quantified using the same method described above (line 597-600). The percentages are then calculated by dividing the number of cells double-positive for GFR α 1/SOX3 and γ H2AX by the number of cells positive for GFR α 1/SOX3.

We have now labelled the cells double-positive for GFR α 1/SOX3 and γ H2AX for clarity (Figure 7D,G) (original Figure 7C, E). There are very few cells that are double-positive for GFR α 1/SOX3 and γ H2AX in control testes at PD14 and PD21. Although there are many cells positive for γ H2AX in control testes, the majority of them are not undifferentiated spermatogonia but meiotic cells, in which γ H2AX signals arise as a result of programmed DNA double strand breaks. On the contrary, γ H2AX is present in most cells positive for GFR α 1/SOX3 in *Brcal* vKO testes at PD14 and PD21, consistent with the quantification data in Figure 7F-I. These observations suggest that DNA damage is not repaired in *Brcal* vKO testes at PD14 and PD21.

- *Can the authors explain why BrdU was chosen in Fig 8A as a marker for replicating cells, whilst in previous Figures, p-H3(S10) was used instead?*

In previous Figures, p-H3(S10), a marker for cells in late G2 to M phases, was used to label proliferating cells. This marker can be easily detected through immunofluorescent staining. BrdU, a marker for cells in S phase, can also be used to label proliferating cells. However, its detection requires additional procedures prior to immunofluorescent staining, including injecting BrdU into mice before testes collection and pre-treating cross-sections with hydrochloric acid (HCl) to denature double-stranded DNA. For this reason, p-H3(S10) was chosen over BrdU to label proliferating cells in previous Figures.

In Figure 8A, we labeled the replicating cells to examine if spontaneous DNA damage arises during DNA replication. BrdU, but not p-H3(S10), is a suitable marker for this purpose.

- *For Fig 8C, the authors used PLZF as a measurement of undifferentiated spermatogonial stem cells, which is inconsistent with the markers with previous quantifications. Similarly, in Fig 8G, the authors used MVH as a measure of germ cells, which is not consistent with previous experiments.*

We agree. To maintain consistency with previous quantifications, we have now used GFR α 1 to measure spermatogonial stem cells, in addition to using PLZF to measure undifferentiated spermatogonia in Figure 8C-D.

As discussed above, we recognize that MVH is not suitable for comparing the number of undifferentiated spermatogonia between control, *Aldh2* KO, *Brcal* vKO, and *Brcal* vKO-*Aldh2* KO mice. To avoid any potential misinterpretation, we have removed the MVH data (original Figure 8G) from the manuscript.

5. One of the main findings of the paper is the unexpected hyper-proliferation of SPG1 in the *Brca1*-vko at PD7 (Figure 4A). This interpretation is based on the relative proportion of the staining of p-H3(S10) vs. GFR α 1 (Figure 4A).

First, based on Fig4A alone it is difficult to be convinced by this because the quality p-H3(S10) stain is not very convincing (i.e. I note the staining is clearer in Fig 7G) and (as point above) it is unclear how many tubules/mice were analysed. This is an important conclusion, and the quantitative data will need to be provided as a table to support Figure 4B.

We have repeated the experiments in Figure 4A and improved the staining of p-H3(S10) and GFR α 1 to match the image quality of Figure 7G. We have also labelled the double-positive cells using arrowheads so that they could be easily identified by readers.

The GFR α 1-positive cells were identified on a single cross-section of testes and their p-H3(S10) status was analyzed. At least 50 GFR α 1-positive cells from each mouse were analyzed. The percentage of p-H3(S10)-positive cells among GFR α 1-positive cells was calculated by dividing the number of cells double-positive for p-H3(S10) and GFR α 1 by the total number of GFR α 1-positive cells analyzed. The percentages obtained from 3 mice were presented. This information is included in the legends of Figure 4. We have included the quantification data used for calculating the relative proportions (Table EV1).

Additionally, data collected across multiple timepoints (PD14 and PD21), shown in the suppl figs 4-5, appear to show reduced undifferentiated spermatogonial stem cells in the mutant mouse model, compared to the controls. My interpretation of the data suggests that "hyper-proliferation" of SPG1 stalls over time. Could you comment?

Although the number of PLZF-positive undifferentiated spermatogonia reduced in *Brca1* vKO testes at PD14 and PD21 compared to controls, the number of GFR α 1-positive spermatogonial stem cells was comparable to controls at PD14 and even increased at PD21. Moreover, we observed that the proliferation rate of GFR α 1-positive spermatogonial stem cells in *Brca1* vKO testes at PD14 and PD21 was still much higher than controls (Appendix Figure S2), indicating that of GFR α 1-positive spermatogonial stem cells keep hyperproliferating from PD7 to PD21.

I was also surprised to see that the number of GFR α 1 cells drops by half between PD7 vs PD14/PD21 in wild-type control.

Undifferentiated spermatogonia consist of two subpopulations: GFR α 1-positive spermatogonial stem cells and SOX3-positive progenitor spermatogonia. During the establishment of undifferentiated spermatogonia in WT mice starting at PD1, GFR α 1-positive spermatogonial stem cells gradually convert into SOX3-positive

progenitor spermatogonia. This process continues until the two subpopulations reach homeostasis at PD14. Since undifferentiated spermatogonia are not fully established by PD7, the number of GFR α 1-positive spermatogonial stem cells continues to decrease from PD7 to PD14, and remains steady thereafter.

Can the authors explain how they have normalised the count of GFR α 1 cells in sup Fig 5B, as there is more cells per 100 μ m of tubules than observed in Fig 4D.

Quantification of GFR α 1-positive cells on cross-sections and in isolated seminiferous tubules was performed using the same method described above (line 597-600, 619-624).

Since immunofluorescent staining was performed on different targets (cross-sections vs. isolated seminiferous tubules), and quantification methods varied (cells per tubule on a 10 μ m cross-section vs. cells per 100 μ m of seminiferous tubule), the numbers of GFR α 1-positive spermatogonial stem cells cannot be directly compared between Figure 4D and Figure EV5B (original supplementary Figure 5B).

As 100 μ m represents a much larger area of the seminiferous tubule than a 10 μ m cross-section, it is not surprising that more cells-and correspondingly, more GFR α 1-positive spermatogonial stem cells-are found in 100 μ m of the seminiferous tubule compared to a 10 μ m cross-section.

Furthermore, there is contradiction in the data shown between Fig 4D and sup Fig 5B, which shows (for the wild-type control) that there is only a decrease in GFR α 1 cells between PD14 and PD21 (sup Fig 5B), whilst the number is halved (as previously stated). This is in contrast to the measurement of SOX3 cells at the same time points between Fig 4F and sup Fig 5D.

Thank you for pointing out this contradiction. We have identified errors in quantifying the number of GFR α 1-positive spermatogonial stem cells at PD14. We have now repeated these experiments and quantified GFR α 1-positive spermatogonial stem cells on cross-section as well as in isolated seminiferous tubules in both control and *Brca1* vKO testes at PD14. The new data for PD14 in Figure 4D and Figure EV5B (original Supplementary Figure 5B) are consistent with each other. We apologize for the confusion caused by our mistakes.

6. In line 259, the authors stated: preventing progenitor spermatogonia's death should restore the homeostasis of undifferentiated spermatogonia and prevent the hyper-proliferation of spermatogonial stem cells. It is not evident earlier in the manuscript if cell death is different wild-type, compared to Brca1 vKO mice. Could you comment

We have now thoroughly analyzed cell death in control and *Brcal* vKO testes at different timepoints (Appendix Figure S3). Little apoptosis was observed in control testes at PD7, PD14, and PD21. In *Brcal* vKO testes, the number of SOX3-positive progenitor spermatogonia was dramatically reduced at PD7, and little apoptosis was observed in the remaining cells. This suggests that the rapid elimination of SOX3-positive progenitor spermatogonia likely prevents us from capturing cells undergoing apoptosis at PD7. Similarly, little apoptosis was observed in GFR α 1-positive spermatogonial stem cells in *Brcal* vKO testes at PD7, likely because the detrimental effects of hyper-proliferation had not yet manifested at this early stage.

By PD14 and PD21, as the number of remaining SOX3-positive progenitor spermatogonia increased in *Brcal* vKO testes, apoptosis became detectable. Concurrently, as the hyper-proliferation of GFR α 1-positive spermatogonial stem cells continued, apoptosis was also observed in this population. Importantly, apoptosis was much more pronounced in SOX3-positive progenitor spermatogonia than in GFR α 1-positive spermatogonial stem cells. These observations are consistent with our previous conclusions that SOX3-positive progenitor spermatogonia are more susceptible to BRCA1 loss than GFR α 1-positive spermatogonial stem cells and the dramatic reduction of progenitor spermatogonia seems to be the most upstream event that disrupts the establishment of homeostasis of undifferentiated spermatogonia.

We have described these observations in the revised manuscript (line 289-305).

Note that it has also been documented that during the first wave of spermatogenesis, spermatogonial stem cells undergo a high frequency of apoptosis (PD8-PD22), compared to adult spermatogonial stem cells (PMID:8989527,20403866,8674410). As such, this might be another possible source of DNA damage.

We have carefully reviewed these three literatures regarding the high frequency of apoptosis in testes during PD8-PD22. However, we did not observe a high frequency of apoptosis at PD14 or PD21 in WT testes in our study (Appendix Figure S3). The causes of the elevated apoptosis during PD8-PD22 in these literatures remain unclear, but they are nevertheless unlikely to be responsible for the DNA damage that occurs during the establishment of undifferentiated spermatogonia, which peaks at PD4 and PD7 (Figure 1C).

*Additionally, in another *Brcal* knock-out mouse model, increased apoptosis in the mutant has been described (PMID:12642502).*

Unlike the complete *Brcal* knock-out mouse model (*Brcal* ^{Δ 5-13/ Δ 5-13}) used in our study, a *Brcal* hypomorphic mouse model (*Brcal* ^{Δ 11/ Δ 11}) was used in the previous study (PMID:12642502), in which most functional domains of BRCA1 are retained. In this *Brcal* hypomorphic mouse model, there are no defects in undifferentiated

spermatogonia, but meiotic cells undergo apoptosis at the pachytene stages of the meiotic prophase due to defective meiotic sex chromosome inactivation (MSCI). Therefore, the increased apoptosis in the *Brca1* hypomorphic mouse model (*Brca1^{AI1/AI1}*) occurs in meiotic cells, but not the undifferentiated spermatogonia as observed in *Brca1* vKO mice in our study.

Minor comments:

I have the following queries regarding the results:

1. Please specify in first section of the results (lines 97 to 130) that the experiments were performed on wild-type control mice.

We have revised the text to specify that these experiments were performed on WT mice (line 106-146).

2. In reference to Figure 2G, there should be a p-value if the authors claim that there is a significant decrease in the percentage of germ cells.

We have used chi-square test to calculate the p-value for the percentage of germ cells in control and *Brca1* vKO samples. The yielding p-value is < 0.00001, suggesting statistical significance. The p-value has been added to Figure 2G.

3. In line 181, the authors stated that the percentage ratio of SPG3 and SPG2 was similar between wild-type control and Brca1 vKO. Is 0.61 (wild-type) 'similar' to 0.98 (Brca1 vKO), as shown in Figure 3C?

We agree that it is not appropriate to use “similar” here. The percentage ratio between SPG3 and SPG2 in *Brca1* vKO testes is 1.6 fold (0.98/0.61) of that in control testes. We have revised the text to precisely describe this result (line 199-203).

4. For supplementary Figure 4A, it is unclear in the Figure legend if the t-SNE plot is in reference to all germ cells (control + Brca1 vKO), or only from control germ cells.

The t-SNE plot is in reference to all germ cells (control + *Brca1* vKO). We have revised the text in the legend of Figure EV4A (original supplementary Figure 4A) accordingly.

5. Figure 4D & 4F: Is the increase observed in Brca1 vKO mice for GFRα1+, or SOX3+, cells significant between PD14 and PD21?

We have performed the statistical analyses, which revealed a significant increase in the numbers of GFRα1-positive and SOX3-positive cell in PD21, compared with those in PD14. The statistical analysis results have been presented in Figure 4D&F, and also in Figure EV5B&D.

6. In Figure 6D, when the number of GFR α 1+ and SOX3+ cells were normalised to percentage of PLZF cells, they appear similar. However, raw SOX3+ cell count per tubules in Figure 6C suggests that there is still a significant reduction between wild-type control and Brca1-p53 vDKO mice. Do these results mean that there is an overall reduced number of PLZF cells in the Brca1-p53 vDKO mouse model? Can the authors show a quantification of PLZF cells in the Brca1-p53 vDKO mouse model compared to wild-type control?

Yes. The quantification revealed that the number of PLZF-positive cells is still fewer in Brca1-p53 vDKO testes compared to control testes at PD7. We have included this data in Figure 6C, and also in Figure 6G. This observation suggests that p53 KO significantly, but not fully, rescues the defects of Brca1 vKO testes. We have described this observation in the revised manuscript (line 320-323).

7. In the same reference to Figure 6D and in regards to lines 268-269, if there is no difference observed for either GFR α 1+, or SOX3+, cells between wild-type control and Brca1-p53 vDKO mice, statistical analyses should be presented in the Figure (i.e. ns).

We have performed the statistical analyses. There is no difference in the percentage of GFR α 1/SOX3-positive cells among PLZF-positive cells between control and Brca1-p53 vDKO mice at PD7 and in the percentage of SOX3-positive cells among PLZF-positive cells between control and Brca1-p53 vDKO mice at PD21. We observed a slight increase (p=0.0438) in the percentages of GFR α 1-positive cells among PLZF-positive cells in Brca1-p53 vDKO mice than control mice at PD21. Results of the statistical analyses have been presented in Figure 6D&H. We have described this observation in the revised manuscript (line 325-329).

8. For the same staining of p-H3(S10) in the tubules, there is a difference in the number of positively stained cells between Figure 7G (top), compared to Figure 4A (top).

We apologize for the confusion caused. The discrepancy might be caused by the low quality of staining in Figure 4A, as you have pointed out above. As suggested, we have repeated the experiments in Figure 4A and improved the staining of p-H3(S10) and GFR α 1 to match the image quality of Figure 7G. The quantification data for the new Figure 4A are similar with those for Figure 7G.

9. The authors hypothesised that increased proliferation with defective DNA repair leads to elevated cell death (lines 309 to 312). The manuscript will be improved if there is supporting evidence of cleaved PARP1 measurement (or a quantitative measurement of cell death) in the seminiferous tubules of their Brca1 vKO mouse model.

We have now thoroughly analyzed cell death in control and *Brcal* vKO testes using cleaved PARP1. As discussed above, the apoptosis is significantly elevated in *Brcal* vKO testes, and the apoptosis was much more pronounced in SOX3-positive progenitor spermatogonia than in GFR α 1-positive spermatogonial stem cells (Appendix Figure S3).

10. *It is not clearly written in lines 321 to 323 that the experiments were performed on wild-type mice. This information was only inferred after looking at Figure 8A.*

We have revised the text to specify that these experiments were performed on WT mice (line 378-380).

11. *In line 329, the authors stated that the BrdU-stained cells are likely Sertoli cells. What is the evidence to support this statement.*

We have now co-stained BrdU and SOX9, a marker for Sertoli cells (Appendix Figure S4) and found around 65% of all BrdU-positive cells are SOX9-positive Sertoli cells in WT testes at PD4 and PD7. This observation supports our previous claim that less than 30% of all BrdU-positive cells are PLZF-positive undifferentiated spermatogonia and the remaining BrdU-positive cells are likely Sertoli cells.

12. *The authors only presented an IF staining for Figure 8G, but have not quantified the number of MVH+ cells (as a separate Figure).*

As discussed above, we recognize that MVH is not suitable for comparing the number of undifferentiated spermatogonia between control, *Aldh2* KO, *Brcal* vKO, and *Brcal* vKO-*Aldh2* KO mice. To avoid any potential misinterpretation, we have removed the MVH data (original Figure 8G) from the manuscript.

13. *Figure 9 (the model) is not referenced in text. Please also add "Timepoints" to the description of this model.*

We have added timepoints (PD1-PD180) to Figure 9 and its legend (line 1055-1066). We have also referenced it in the text (line 447).

Typographical errors and corrections

14. *In general, authors should list the number of replicates in Figure legend.*

We have now included the number of replicates in the legends for all Figures.

Typo in Figure legend of supplementary Figure 4E. It should be SOX3+, instead of GFR α 1+.

This typographical error has been corrected in the Figure legend of Figure EV4D (original supplementary Figure 4E).

15. In Figure 5E, a scale bar is missing for control PD90 image.

The scale bar has been added in Figure 5D (original Figure 5E).

16. Check that murine 'BRCA1' is written as *Brca1* throughout the text

We have kept the protein symbol as “BRCA1” (all uppercase letters) in the manuscript according to the most recent guidelines issued by International Committee on Standardized Genetic Nomenclature for Mice and Rat Genome and Nomenclature Committee (<https://www.informatics.jax.org/mgihome/nomen/gene.shtml>, revised September, 2024), which suggest that “Protein symbols use all uppercase letters”.

Referee #2:

This manuscript, authored by Li et al., investigates the impact of germline-specific conditional knockout of BRCA1 on the cellular dynamics of spermatogonia in mice. BRCA1, an ubiquitin ligase, is involved in various DNA repair pathways. Previous studies have shown that complete BRCA1 knockout leads to embryonic lethality, and truncated hypomorphic alleles, conditional knockouts, and compound mutations (e.g., with p53) have revealed the role of BRCA1 in spermatogenesis, ovarian reserve, and meiosis.

In this study, the authors re-examine spermatogenesis in male mice with a germline-specific Brca1 knockout, focusing on the number of spermatogonial stem cells (SSCs) and progenitor spermatogonia using detailed histological analysis. A key finding is that Brca1 loss-of-function in the male germline results in a more severe reduction in the number of progenitor spermatogonia compared to SSCs. Another observation is that SSCs in neonatal testes transiently exhibit strong gH2AX staining, which is absent in adult testes. The authors suggest that gH2AX staining indicates higher levels of DNA damage in neonatal SSCs. However, alternative interpretations of gH2AX staining should be considered. Additionally, the authors propose that formaldehyde, endogenously generated in neonatal SSCs, may contribute to DNA damage, based on the observation that the number of spermatogonia in the Brca1 and Aldh2 double knockout is reduced more severely than in the single knockouts. This interpretation again requires more careful consideration of the data.

While the manuscript provides potentially interesting insights into the regulation of genome integrity in SSCs, the causal relationships among the three main findings (1) the reduced number of neonatal SSCs in Brca1 knockout, (2) the increased gH2AX staining in neonatal SSCs, and (3) the proposed role of formaldehyde, are not

experimentally clear. Additional data are needed to support the authors' hypothesis. My major points are as follows:

Thank you for raising these important points, which have significantly contributed to improving our manuscript. In response, we have conducted additional experiments and analyses, and we have thoroughly revised the manuscript to address these concerns.

Major Points:

1. Interpretation of γ H2AX signals

γ H2AX signals increase not only in response to DNA double-strand breaks (DSBs) but also due to chromatin conformation changes. Actually, drastic epigenetic changes should occur in neonatal spermatogonial stem cells. To distinguish between DSBs and chromatin changes, additional data on DSBs are needed. This could include immunostaining for other DNA damage markers (e.g., 53BP1), the comet assay, or DNA damage sequencing.

We agree that besides DNA damage, chromatin conformation changes could also induce γ H2AX signals. To examine the cause for γ H2AX signals, we carefully examined the patterns of γ H2AX signals in undifferentiated spermatogonia in WT testes at PD4 and PD7. We observed γ H2AX foci in around 50% of undifferentiated spermatogonia and pan-nuclear γ H2AX signals in around 35% of undifferentiated spermatogonia (Figure EV2A-B).

Comet assay and DNA damage sequencing are useful for detecting both single-stranded and double-stranded DNA breaks. We apologize for not being able to perform these experiments due to technical challenges in purifying undifferentiated spermatogonia and establishing the DNA damage sequencing methods in the lab. As suggested, we performed immunostaining for another DNA damage marker 53BP1, which specifically localizes to DSBs. However, we could not reliably identify cells with 53BP1 foci in undifferentiated spermatogonia at PD4 and PD7. There are two possible explanations for this observation: 1) This 53BP1 antibody works well for immunostaining on cell culture but not on testis cross-sections. 2) The γ H2AX foci in undifferentiated spermatogonia do not represent DSBs but other types of DNA damage that is unclear yet.

Previous cellular studies have demonstrated that γ H2AX can appear as either distinct foci or pan-nuclear signals in response to DNA damage, depending on the DNA damaging agent used. On the contrary, chromatin conformation changes usually leads to pan-nuclear γ H2AX signals. Consistent with previous findings, we observed pan-nuclear γ H2AX signals in mouse embryonic fibroblasts treated with Panobinostat (LBH589), an HDAC inhibitor that induces chromatin relaxation (Figure R1). We also observed various patterns of γ H2AX signals in mouse embryonic fibroblasts treated with different DNA damaging agents: pan-nuclear γ H2AX signals in cells treated with

hydroxyurea, γ H2AX foci in cells treated with bleomycin, and a mixture of γ H2AX foci and pan-nuclear γ H2AX signals in cells treated with acetaldehyde (Figure R1). Therefore, the patterns of γ H2AX signals (a mixture of H2AX foci and pan-nuclear γ H2AX signals) are more consistent with the presence of DNA damage in undifferentiated spermatogonia.

In addition, the γ H2AX signals in undifferentiated spermatogonia diminish in WT testes but not *Brcal* vKO testes at PD14 and PD21. Since BRCA1 has a well-characterized role in DNA damage repair but not in epigenetic regulation or chromatin remodeling, the phenotype in *Brcal* vKO testes is best explained by DNA damage repair deficiency. Taken together, although we failed to reliably identify cells with 53BP1 foci in undifferentiated spermatogonia, the patterns of γ H2AX signals and their retention in *Brcal* vKO testes suggest that DNA damage is more likely than chromatin conformation changes to be the primary trigger of γ H2AX signals in undifferentiated spermatogonia.

We agree that drastic epigenetic changes should occur during the formation of undifferentiated spermatogonia. Although DNA damage is likely the primary trigger for the γ H2AX signals, we could not formally exclude the possibility that drastic epigenetic changes could trigger global chromatin conformation changes, which contribute in part to the pan-nuclear γ H2AX signals. We have discussed this possibility in the revised manuscript (line 470-473). During the drastic epigenetic changes, formaldehyde could arise from histone demethylation and generate DNA damage, making it a potential link between drastic epigenetic changes and the γ H2AX signals. We have also discussed this possibility in the revised manuscript (line 453-469).

Figure for referees not shown.

2. Quantification of γ H2AX foci

gH2AX typically forms nuclear foci, and the number of these foci is commonly quantified to assess DNA damage. It is unclear why the standard quantification of gH2AX foci was not performed in this study. If BRCA1 functions to repair DNA damage detected by gH2AX in spermatogonia, one would expect the signals or foci of gH2AX (and other DNA damage markers) to be higher in Brca1 KO testes compared to controls. However, the authors did not present such data. More careful experiments are needed to support the claim that "BRCA1 preserves genome integrity during the formation of undifferentiated spermatogonia," as stated in the manuscript's title.

We agree that the quantification of γ H2AX signals or foci is a common method for assessing DNA damage in cells. Since both γ H2AX foci and pan-nuclear γ H2AX signals were observed in undifferentiated spermatogonia, quantifying γ H2AX foci alone could not reflect the overall γ H2AX levels in undifferentiated spermatogonia. Therefore, instead of quantifying γ H2AX foci alone, we quantified the intensities of γ H2AX signals (both γ H2AX foci and pan-nuclear γ H2AX signals) in undifferentiated spermatogonia. To obtain an additional parameters for evaluation, we also calculated the percentages of γ H2AX-positive undifferentiated spermatogonia (either γ H2AX foci or pan-nuclear γ H2AX signals) among undifferentiated spermatogonia.

At PD7, the intensities of γ H2AX signals and the percentages of γ H2AX-positive undifferentiated spermatogonia were similar between control and *Brca1* vKO testes (Figure 7A-C). Since DNA damage are not repaired in either testes at this timepoint, the intensities of γ H2AX signals are determined by the amount of endogenous DNA damage generated and are unlikely affected the DNA damage repair deficiency caused by BRCA1 loss. At PD14 and PD21, the intensities of γ H2AX signals and the percentages of γ H2AX-positive undifferentiated spermatogonia were very low in control testes at PD14 and PD21 (Figure 7D-I), suggesting that DNA damage repair is largely completed. In contrast, the intensities of γ H2AX signals and the percentages of γ H2AX-positive undifferentiated spermatogonia remained high in *Brca1* vKO testes at PD14 and PD21 (Figure 7D-I), indicating that DNA damage repair is significantly delayed, consistent with DNA damage repair deficiency caused by BRCA1 loss. These findings suggest that BRCA1 is essential for efficient DNA damage repair, which is critical for maintaining genome integrity during the formation of undifferentiated spermatogonia.

3. Differentiation of spermatogonia

It is known that the differentiation process of spermatogonia during the first wave of spermatogenesis in neonatal testes is distinct from that in adult testes. At postnatal days 4-5 (PD4-5), spermatogonial stem cells likely begin to differentiate. Therefore, a potential correlation between spermatogonial differentiation (toward differentiating spermatogonia and meiosis) and increased gH2AX signals in neonatal testes should be carefully examined and discussed.

This is indeed an important point that we should have tested. To examine a potential correlation between spermatogonial differentiation and increased γ H2AX signals in neonatal testes, we co-stained γ H2AX and KIT, a well-characterized marker for differentiating spermatogonia, in testis sections at PD4. Our results show that only around 20% of γ H2AX-positive cells are KIT-positive (Figure EV2E-F). Since most γ H2AX-positive cells are KIT-negative, it is unlikely that γ H2AX signals in neonatal testes are caused by spermatogonial differentiation. Instead, our data suggest that DNA damage occurs during the establishment of undifferentiated spermatogonia and is not fully repaired before spermatogonial differentiation begins. This is consistent with the persistence of γ H2AX signals in neonatal testes until PD14. Furthermore, although spermatogonial differentiation continues throughout the reproductive lifespan, γ H2AX signals do not reappear after PD14, which further supports the idea that spermatogonial differentiation is not the cause of γ H2AX signals. We have discussed these findings in the revised manuscript (line 136-140).

4. Clarification of DNA Damage generation and repair

In line 304, the authors state, "since DNA damage generation and repair deficiency were similar between these two populations in Brca1 vKO testes." However, how can DNA damage generation and repair be inferred solely from γ H2AX staining? Further clarification or additional experiments are needed to support this claim.

Thank you for pointing this out. We apologize for the confusion caused by our incorrect statement. We have removed this sentence and revised the text in the manuscript (line 363-369). Below is the clarification of what we intended to convey about the DNA damage repair status in GFR α 1-positive spermatogonial stem cells and SOX3-positive progenitor spermatogonia.

In WT testes, the γ H2AX signals arise similarly in both populations at PD7 and diminish in both populations by PD14, suggesting that the DNA damage repair dynamics are similar between these two populations in WT testes. In *Brca1* vKO testes, BRCA1 is lost in both populations, and strong γ H2AX signals persist in both populations at PD14 and PD21, indicating that DNA damage repair is similarly impaired in both populations in *Brca1* vKO testes.

5. Proliferation vs. cell death

In line 309, the authors state, "since cell proliferation in the presence of unrepaired DNA damage is detrimental, the higher proliferation rate may lead to elevated cell death in progenitor spermatogonia when spontaneous DNA damage fails to be repaired efficiently in the absence of BRCA1." I agree with the general concept that highly proliferating cells are more susceptible to DNA damage. However, if this is true, why do the hyperproliferating spermatogonial stem cells in Brca1 KO not undergo more cell death, but instead continue to hyperproliferate? This discrepancy should be addressed.

We apologize for the confusion caused. Our intention was not to suggest that the hyper-proliferating spermatogonial stem cells do not undergo more cell death but instead continue to hyper-proliferate in *Brca1* vKO testes. In fact, this hyper-proliferation is transient. While the number of spermatogonial stem cells significantly increases at PD14 and PD21, it subsequently declines and eventually becomes exhausted by PD180 (Figure 4D, Figure 5B), indicating that increased cell death occurs following this transient hyper-proliferation. Consistently, we observed a dramatic increase in apoptosis among spermatogonial stem cells in *Brca1* vKO testes at PD14 and PD21 (Appendix Figure S3). These findings suggest that hyper-proliferation is detrimental to spermatogonial stem cells in *Brca1* vKO testes. This observation aligns with the idea that an increased proliferation rate enhances susceptibility to DNA damage, ultimately leading to cell death. We have clarified this point in the revised manuscript (line 286-305).

6. Lack of cell death analysis

While the manuscript includes several studies on cell proliferation kinetics, only a single panel (Figure 1D) is presented for cell death. A more thorough investigation of cell death, which is crucial for interpreting the data, would strengthen the manuscript.

We have now thoroughly analyzed cell death in control and *Brca1* vKO testes at different timepoints (Appendix Figure S3). Little apoptosis was observed in control testes at PD7, PD14, and PD21. In *Brca1* vKO testes, the number of SOX3-positive progenitor spermatogonia was dramatically reduced at PD7, and little apoptosis was observed in the remaining cells. This suggests that the rapid elimination of SOX3-positive progenitor spermatogonia likely prevents us from capturing cells undergoing apoptosis at PD7. Similarly, little apoptosis was observed in GFR α 1-positive spermatogonial stem cells in *Brca1* vKO testes at PD7, likely because the detrimental effects of hyper-proliferation had not yet manifested at this early stage.

By PD14 and PD21, as the number of remaining SOX3-positive progenitor spermatogonia increased in *Brca1* vKO testes, apoptosis became detectable. Concurrently, as the hyper-proliferation of GFR α 1-positive spermatogonial stem cells continued, apoptosis was also observed in this population. Importantly, apoptosis was much more pronounced in SOX3-positive progenitor spermatogonia than in GFR α 1-positive spermatogonial stem cells. These observations are consistent with our previous conclusions that SOX3-positive progenitor spermatogonia are more susceptible to BRCA1 loss than GFR α 1-positive spermatogonial stem cells and the dramatic reduction of progenitor spermatogonia seems to be the most upstream event that disrupts the establishment of homeostasis of undifferentiated spermatogonia.

We have described these observations in the revised manuscript (line 289-305).

7. Single cell RNA sequencing (scRNAseq) data

The authors use scRNAseq to quantify the number and proportion of spermatogonia (SPG1-3). However, no mention is made of potential differences in gene expression, such as those downstream of the DNA damage checkpoint or other relevant pathways. This should be addressed.

We agree that it is important to analyze if differential gene expression contributes to the phenotypes observed in *Brca1* vKO testes. As suggested, we conducted differential expression gene analysis for the SPG1, SPG2, and SPG3 clusters and then performed pathway enrichment analysis on the up- and down-regulated genes using the DAVID website. The top ten enriched pathways with p-value less than 0.05 are presented in Appendix Figure S1.

Our analysis of scRNAseq data has revealed that the germ cell development from SPG1 to SPG2 was most affected in *Brca1* vKO testes, and SPG2 was much more susceptible to BRCA1 loss than SPG1, and the susceptibility was not further increased in SPG3 population (Figure 3C). If the potential differences in gene expression directly contribute to the changes in spermatogonia population phenotype, these expression difference are more likely to be identified in SPG1 (spermatogonial stem cells) and SPG2 (progenitor spermatogonia) clusters. In contrast, gene expression changes in the SPG3 are more likely to represent consequences rather than causative factors. In this case, we paid particular attention to the pathways enriched in the SPG1 and SPG2 populations. Based on our analysis, we did not identify DNA damage checkpoint or relevant pathways in the SPG1 and SPG2 enrichment output. This suggests that gene expression changes are unlikely to contribute directly to the phenotypes in *Brca1* vKO testes.

8. *Brca1* and *Aldh2* double knockout

*The authors present data showing that the number of spermatogonia is more severely reduced in *Brca1* and *Aldh2* double knockout testes compared to each single knockout. However, γ H2AX and other DNA damage markers (including signal intensities and nuclear foci) are not examined. It is unclear whether "formaldehyde is likely a source of spontaneous DNA damage during the formation of undifferentiated spermatogonia" can be concluded from simply counting the number of cells. Further analysis (to establish cause and effect) of DNA damage in the double knockout model would be necessary to support this hypothesis.*

We agree that it is important to analyze the DNA damage status in *Brca1* vKO-*Aldh2* KO mice. The intensities of γ H2AX signals and the percentages of γ H2AX-positive undifferentiated spermatogonia remained high in both *Brca1* vKO and *Brca1* vKO-*Aldh2* KO testes at PD21 (Appendix Figure S5C-D), indicating that DNA damage repair is significantly delayed in both testes, consistent with DNA damage repair deficiency caused by BRCA1 loss.

Although we predict that more DNA damage is generated in the undifferentiated spermatogonia of *Brcal* vKO-*Aldh2* KO testes, there is no significant difference in the intensities of γ H2AX signals and the percentages of γ H2AX-positive undifferentiated spermatogonia between *Brcal* vKO and *Brcal* vKO-*Aldh2* KO testes (Appendix Figure S5C-D). Since strong γ H2AX signals are already present in *Brcal* vKO testes, the potential increase in DNA damage in *Brcal* vKO-*Aldh2* KO might not be able to be detected by immunofluorescent staining, a semi-quantitative method.

We fully acknowledge that the only difference we have observed is a more severe reduction of undifferentiated spermatogonia in *Brcal* vKO-*Aldh2* KO testes. We have added caveats in the revised text to acknowledge this important limitation (line 432). We have also downplayed our claims about formaldehyde as a source of DNA damage in undifferentiated spermatogonia (line 371-372, 430-431).

Referee #3:

This is an interesting paper creating insights into the establishment of spermatogonial stem cell populations. The paper uses elegant and up-to-date technologies and presents valid datasets.

The authors report somewhat surprising findings on DNA damage of spermatogonial subpopulations by thus far neglected mechanisms.

The paper provides significant and thoughtful new insights. I consider the discussion too long and slightly unfocused.

Otherwise the studies are well performed, the paper is carefully designed and the data are well presented.

Thank you for your positive feedback on our study. We have revised the discussion to make it more concise and focused.

References:

Dura M, Teissandier A, Armand M, Barau J, Lapoujade C, Fouchet P, Bonneville L, Schulz M, Weber M, Baudrin LG *et al* (2022) DNMT3A-dependent DNA methylation is required for spermatogonial stem cells to commit to spermatogenesis. *Nat Genet* 54: 469-480

Wang Z, Xu X, Li JL, Palmer C, Maric D, Dean J (2019) Sertoli cell-only phenotype and scRNA-seq define PRAMEF12 as a factor essential for spermatogenesis in mice. *Nat Commun* 10: 5196

Dear Dr. Liu,

Thank you for the submission of your revised manuscript to our editorial offices. I have now received the reports from one of the two the referees (Referee #1) that were asked to re-evaluate the study, you will find below. Referee #2 was completely unresponsive to our invitations to re-assess the study. However, going through your p-b-p-response, I consider his/her concerns as adequately addressed. Referee #3 was basically already happy with the previous version of the manuscript and was thus not consulted again.

As you will see, referee #1 now supports the publication of your manuscript in EMBO reports. However, s/he has remaining concerns and suggestions to improve the manuscript, I ask you to address in a final revised manuscript. Please also the attached detailed response to your rebuttal by referee #1. Please provide a final p-b-p-response to the remaining issues and the editorial requests below.

I have these further editorial requests:

- We now use CRediT to specify the contributions of each author in the journal submission system. CRediT replaces the author contribution section. Please use the free text box to provide more detailed descriptions and do NOT provide your final manuscript text file with an author contributions section. See also our guide to authors:
<https://www.embopress.org/page/journal/14693178/authorguide#authorshippinguidelines>

- Please check again that the number "n" for how many independent experiments were performed, their nature (biological versus technical replicates), the bars and error bars (e.g. SEM, SD) and the test used to calculate p-values is indicated in the respective figure legends. Please also check that all the p-values are explained in the legend, and that these fit to those shown in the figure. Please provide statistical testing where applicable. Please avoid the phrase 'independent experiment' but clearly state if these were biological or technical replicates. Please also indicate (e.g. with n.s.) if testing was performed, but the differences are not significant. In case n=2, please show the data as separate datapoints without error bars and statistics. See also:
<http://www.embopress.org/page/journal/14693178/authorguide#statisticalanalysis>

If n<5, please show single datapoints for diagrams. It seems presently many diagrams are missing the 'n.s.'. Moreover:

- Please add to each legend (main and EV figures, where applicable) a 'Data Information' section (or name the provided section like this) explaining the statistics used or providing information regarding replicates and scales. See:

- Please remove the 'Reagents and Tools Table' from the Methods section of the main manuscript text file. This should only be uploaded separately.

- There is a Table EV1 uploaded and called out. This is a dataset. Please upload this as dataset file named Dataset EV1, add a legend on the first TAB and update the callout accordingly.

- Please remove now the referee token from The Data Availability section (DAS) and make sure the dataset is public latest on the date of online publication of the study.

- During our figure integrity check, we noted a partial overlap between Figure 2H (lower left image, control, PD7) and Figure 3D (upper right image, control, PLZF MVH). Please check. If the reuse is intentional, please indicate this clearly in the respective figure legend.

In addition, I would need from you uploaded separately:

- a short, two-sentence summary of the manuscript (not more than 35 words).

- two to four short (!) bullet points highlighting the key findings of your study (two lines each).

- a schematic summary figure as separate file that provides a sketch of the major findings (not a data image) in jpeg or tiff format (with the exact width of 550 pixels and a height of not more than 400 pixels) that can be used as a visual synopsis on our website.

I look forward to seeing a new revised version of your manuscript as soon as possible.

Best,

Achim Breiling
Editor
EMBO Reports

Referee #1:

The manuscript has been improved and the flow and choice of timepoints have been clarified. As a general comment, the authors have only provided a single manuscript file and have not detailed the changes implemented in their rebuttal. Moreover, the line numbers of the new version do not match those listed in the rebuttal. It is nearly impossible for this reviewer to assess the changes and what has been implemented in the new version. It would have been helpful if changes in the new version of the manuscript had been highlighted, as text in a different colour, or described in detail in the rebuttal.

Regarding point 3 and the scRNAseq data: We understand that it is not possible to re-do these experiments and they add value to the study. A single biological sample is particularly problematic because the interpretation of the data is based on relative proportion of cell clusters. We would suggest that the authors mention the limitations of their scRNA-seq themselves in the text, as this is not best practice and explain to the reader how they validate their data using immunofluorescence at different developmental stages.

Likewise in response to the comment about testicular volume: the authors could address this in the methods, explaining that they input more of the Brca1 vKO in order to compensate for the known smaller testicular volumes of the mutant.

Regarding point 4 and MVH staining: Whilst the authors have removed MVH as a staining for undifferentiated spermatogonial stem cells in the original Figure 5D, they have not addressed the question of using MVH as a marker in Figure 2H and Figure 3D. This is an inconsistency in the choice for a marker to use MVH, whilst for other staining, PLZF was used as a marker. Can the authors address why MVH was used in these two figures, or show quantification of cells with PLZF instead?

Additional comments:

Line 35: Typo with the additional word (underlined): BRCA1 loss leads to in a dramatic reduction...

Line 87-88: "No defects in undifferentiated spermatogonia have been reported in Brca1 hypomorphic mice" - this sentence should be corrected, as it is not the mouse that is hypomorphic, instead, it is the Brca1 allele.

Lines 95-98: " Given that undifferentiated spermatogonia are formed within a short time frame, endogenous DNA damage accumulated during this stage might be expected to be too low to threaten genomic stability". I struggle to understand the meaning or rationale of this sentence - can you please explain. Undifferentiated spermatogonia are first specified during the first wave of spermatogenesis but as they self-renew, they are 'formed' during the whole duration of fertile life?

Line 98-99: "... during the formation of undifferentiated spermatogonia" - again as above, it is unclear what the 'formation' of undifferentiated spermatogonia refers to? I note that the description is very vague about exact timing of the events described in this summary paragraph.

Line 141: typo with the additional word (underlined): ... unlikely the reason for the γ H2AX signals in in PLZF-positive germ cells.

Lines 232-235: The authors will need to provide a clearer explanation in the discussion as to why gene expression changes are unlikely to contribute to the phenotype in Brca1 vKO mice from SPG1 to SPG2. This statement in the results contradicts the discussion, where the authors speculate that massive transcriptional reprogramming occurs during the formation of undifferentiated spermatogonia (lines 464-466 and 470-473).

Referee #1:

The manuscript has been improved and the flow and choice of timepoints have been clarified.

Thank you for your positive feedback on our revised manuscript.

As a general comment, the authors have only provided a single manuscript file and have not detailed the changes implemented in their rebuttal. Moreover, the line numbers of the new version do not match those listed in the rebuttal. It is nearly impossible for this reviewer to assess the changes and what has been implemented in the new version. It would have been helpful if changes in the new version of the manuscript had been highlighted, as text in a different colour, or described in detail in the rebuttal.

In the previous revision, the line numbers listed in the rebuttal match those in the manuscript file (word format) that we have uploaded. However, the line numbers were altered in the manuscript file after being converted to the PDF format by the submission system. We apologize for the inconvenience caused. In this revision, the line numbers listed in the rebuttal match those in the manuscript file (PDF format).

Regarding point 3 and the scRNAseq data: We understand that it is not possible to re-do these experiments and they add value to the study. A single biological sample is particularly problematic because the interpretation of the data is based on relative proportion of cell clusters. We would suggest that the authors mention the limitations of their scRNA-seq themselves in the text, as this is not best practice and explain to the reader how they validate their data using immunofluorescence at different developmental stages.

We have now included the limitations of our scRNA-seq experiments. We have also explained that the scRNA-seq data were validated using immunofluorescence at different developmental stages (line 173-176).

Likewise in response to the comment about testicular volume: the authors could address this in the methods, explaining that they input more of the Brca1 vKO in order to compensate for the known smaller testicular volumes of the mutant.

We have now included this information in the methods (line 635-637).

Regarding point 4 and MVH staining: Whilst the authors have removed MVH as a staining for undifferentiated spermatogonial stem cells in the original Figure 5D, they have not addressed the question of using MVH as a marker in Figure 2H and Figure 3D. This is an inconsistency in the choice for a marker to use MVH, whilst for other staining, PLZF was used as a marker. Can the authors address why MVH was used in

these two figures, or show quantification of cells with PLZF instead?

Our initial analysis of scRNA-seq data revealed that the percentage of germ cell population significantly decreased in *Brca1* vKO testes at PD7 (Figure 3G). To validate this observation, we performed immunofluorescent staining using germ cell marker MVH (Figure 2H). Our further analysis of germ cell populations in scRNA-seq data revealed that the formation of undifferentiated spermatogonia was defective in *Brca1* vKO testes at PD7 (Figure 3A-C). To validate this observation, we first performed immunofluorescent staining to confirm the identity of the remaining germ cells (using germ cell marker MVH) in *Brca1* vKO testes at PD7 (Figure 3D). Since most remaining MVH-positive germ cells in *Brca1* vKO testes were PLZF-positive undifferentiated spermatogonia at PD7 (Figure 3D), we used PLZF for immunofluorescent staining in the subsequent analyses of the two populations of undifferentiated spermatogonia.

Additional comments:

Line 35: Typo with the additional word (underlined): BRCA1 loss leads to in a dramatic reduction...

This typographical error has been corrected (line 35).

Line 87-88: "No defects in undifferentiated spermatogonia have been reported in Brca1 hypomorphic mice" - this sentence should be corrected, as it is not the mouse that is hypomorphic, instead, it is the Brca1 allele.

We agree. We have corrected this sentence according to your suggestions (line 86-87).

Lines 95-98: " Given that undifferentiated spermatogonia are formed within a short time frame, endogenous DNA damage accumulated during this stage might be expected to be too low to threaten genomic stability". I struggle to understand the meaning or rational of this sentence - can you please explain. Undifferentiated spermatogonia are first specified during the first wave of spermatogenesis but as they self-renew, they are 'formed' during the whole duration of fertile life?

Sorry for the confusions caused. In our study, “the formation of undifferentiated spermatogonia” refers to the conversion of spermatogonial stem cells to progenitor spermatogonia until the homeostasis between these two populations are established, which occur shortly after birth. Once the undifferentiated spermatogonia are formed, the homeostasis between these two populations are maintained to support long-term male fertility. We have now clarified this idea (line 93-97).

Line 98-99: "... during the formation of undifferentiated spermatogonia" - again as above, it is unclear what the 'formation' of undifferentiated spermatogonia refers to? I

note that the description is very vague about exact timing of the events described in this summary paragraph.

As mentioned above, “the formation of undifferentiated spermatogonia” refers to the conversion of spermatogonial stem cells to progenitor spermatogonia until the homeostasis between these two populations are established, which occur shortly after birth. The exact timing of the events (PD1-PD14) were not clear until we have carefully examined in this study (Figure EV1).

Line 141: typo with the additional word (underlined): ... unlikely the reason for the γ H2AX signals in in PLZF-positive germ cells.

This typographical error has been corrected (line 141).

*Lines 232-235: The authors will need to provide a clearer explanation in the discussion as to why gene expression changes are unlikely to contribute to the phenotype in *Brcal* vKO mice from SPG1 to SPG2. This statement in the results contradicts the discussion, where the authors speculate that massive transcriptional reprogramming occurs during the formation of undifferentiated spermatogonia (lines 464-466 and 470-473).*

Sorry for the confusions caused. The “gene expression changes” refers to those caused by BRCA1 loss, which are different from the massive transcriptional reprogramming that occurs during the formation of undifferentiated spermatogonia in both WT and *Brcal* vKO mice. We speculate that massive transcription reprogramming leads to a surge in the level of formaldehyde, a potential source of spontaneous DNA damage that arises during the formation of undifferentiated spermatogonia in both WT and *Brcal* vKO mice.

During the analysis of gene expression changes between WT and *Brcal* vKO mice, we did not identify the enrichment of DNA damage checkpoint or relevant pathways in SPG1 or SPG2. Therefore, the gene expression changes caused by BRCA1 loss are unlikely to contribute directly to the phenotypes in *Brcal* vKO testes. We have modified this sentence to clarify this idea (line 228-236).

Editorial requests:

- We now use CRediT to specify the contributions of each author in the journal submission system. CRediT replaces the author contribution section. Please use the free text box to provide more detailed descriptions and do NOT provide your final manuscript text file with an author contributions section. See also our guide to authors:

<https://www.embopress.org/page/journal/14693178/authorguide#authorshipguidelines>

Done.

- Please check again that the number "n" for how many independent experiments were performed, their nature (biological versus technical replicates), the bars and error bars (e.g. SEM, SD) and the test used to calculate p-values is indicated in the respective figure legends. Please also check that all the p-values are explained in the legend, and that these fit to those shown in the figure. Please provide statistical testing where applicable. Please avoid the phrase 'independent experiment' but clearly state if these were biological or technical replicates. Please also indicate (e.g. with n.s.) if testing was performed, but the differences are not significant. In case n=2, please show the data as separate datapoints without error bars and statistics. See also:

<http://www.embopress.org/page/journal/14693178/authorguide#statisticalanalysis>

If $n < 5$, please show single datapoints for diagrams. It seems presently many diagrams are missing the 'n.s.'.

Done.

Moreover:

- Please add to each legend (main and EV figures, where applicable) a 'Data Information' section (or name the provided section like this) explaining the statistics used or providing information regarding replicates and scales. See:

Done.

- Please remove the 'Reagents and Tools Table' from the Methods section of the main manuscript text file. This should only be uploaded separately.

Done.

- There is a Table EV1 uploaded and called out. This is a dataset. Please upload this as dataset file named Dataset EV1, add a legend on the first TAB and update the

callout accordingly.

Done.

- Please remove now the referee token from The Data Availability section (DAS) and make sure the dataset is public latest on the date of online publication of the study.

Done.

- During our figure integrity check, we noted a partial overlap between Figure 2H (lower left image, control, PD7) and Figure 3D (upper right image, control, PLZF MVH). Please check. If the reuse is intentional, please indicate this clearly in the respective figure legend.

Thank you for pointing this out. Figure 2H (control, PD7, MVH) and Figure 3D (control, PD7, PLZF+MVH) were indeed from the same section of PD7 control mice stained with PLZF and MVH. However, we do not intend to reuse the figures. To avoid any confusions, we have updated Figure 2H (control, PD7, MVH) with a figure from a different section of PD7 control mice stained with MVH.

Yidan Liu
Zhejiang University
Hangzhou, Zhejiang 310000
China

Dear Dr. Liu,

I am very pleased to accept your manuscript for publication in the next available issue of EMBO reports. Thank you for your contribution to our journal.

Yours sincerely,
